# Normoglycemia and physiological cortisone level maintain glucose homeostasis in a pancreas-liver microphysiological system
Sophie Rigal[1], Belén Casas[2,3], Kajsa P. Kanebratt[2], Charlotte Wennberg Huldt[4], Lisa U. Magnusson[5], Erik Müllers [5], Fredrik Karlsson[6], Maryam Clausen[7], Sara F. Hansson[8], Louise Leonard[6], Jonathan Cairns [9], Rasmus Jansson Löfmark[2], Carina Ämmälä[4], Uwe Marx[1], Peter Gennemark [2,3], Gunnar Cedersund[3,10], Tommy B. Andersson[2] & Liisa K. Vilén [2,11] ✉

Current research on metabolic disorders and diabetes relies on animal models because multi-organ diseases cannot be well studied with standard in vitro assays. Here, we have connected cell models of key metabolic organs, the pancreas and liver, on a microfluidic chip to enable diabetes research in a human-based in vitro system. Aided by mechanistic mathematical modeling, we demonstrate that hyperglycemia and high cortisone concentration induce glucose dysregulation in the pancreas-liver microphysiological system (MPS), mimicking a diabetic phenotype seen in patients with glucocorticoid-induced diabetes. In this diseased condition, the pancreas-liver MPS displays beta-cell dysfunction, steatosis, elevated ketone-body secretion, increased glycogen storage, and upregulated gluconeogenic gene expression. Conversely, a physiological culture condition maintains glucose tolerance and beta-cell function. This method was reproducible in two laboratories and was effective in multiple pancreatic islet donors. The model also provides a platform to identify new therapeutic proteins, as demonstrated with a combined transcriptome and proteome analysis.

Glucose homeostasis, a tightly regulated process, maintains blood glucose levels within a narrow range. Dysfunctional regulation leads to severe diseases like diabetes mellitus, characterized by hyperglycemia due to impaired inter-organ communication. Understanding the mechanisms of glucose dysregulation is crucial for developing new effective treatments. In healthy individuals, pancreatic beta cells respond to increased blood glucose concentration by secreting insulin (Fig. 1a), which in turn regulates glucose uptake and storage in the liver and other target organs[1]. The liver plays a central role in glucose homeostasis by storing glucose as glycogen or lipids during hyperglycemia and producing glucose via gluconeogenesis during hypoglycemia to balance blood glucose levels[2]. Glucose dysregulation occurs when the target organs become resistant to insulin, leading to improper blood glucose control (Fig. 1b). Insulin resistance, in turn, evokes increased insulin secretion (beta-cell adaptation) to compensate for impaired insulin sensitivity (Fig. 1b), and may ultimately lead to pancreatic beta-cell failure and type 2 diabetes[3] (Fig. 1c). Furthermore, certain drug treatments, such as

¹TissUse GmbH, Berlin, Germany. ²Drug Metabolism and Pharmacokinetics, Research and Early Development, Cardiovascular, Renal and Metabolism (CVRM), BioPharmaceuticals R&D, AstraZeneca, Gothenburg, Sweden. ³Department of Biomedical Engineering, Linköping University, Linköping, Sweden. ⁴Bioscience Metabolism, Research and Early Development, Cardiovascular, Renal and Metabolism (CVRM), BioPharmaceuticals R&D, AstraZeneca, Gothenburg, Sweden. ⁵Bioscience Cardiovascular, Research and Early Development, Cardiovascular, Renal and Metabolism (CVRM), BioPharmaceuticals R&D, AstraZeneca, Gothenburg, Sweden. ⁶Data Sciences and Quantitative Biology, Discovery Sciences, R&D, AstraZeneca, Gothenburg, Sweden. ⁷Translational Genomics, Discovery Biology, Discovery Sciences, R&D, AstraZeneca, Gothenburg, Sweden. ⁸Translational Science and Experimental Medicine, Research and Early Development, Cardiovascular, Renal and Metabolism (CVRM), BioPharmaceuticals R&D, AstraZeneca, Gothenburg, Sweden. ⁹Data Sciences and Quantitative Biology, Discovery Sciences, BioPharmaceuticals R&D, AstraZeneca, Cambridge, UK. ¹⁰Center for Medical Image Science and Visualization (CMIV), Linköping University, Linköping, Sweden. ¹¹Division of Pharmaceutical Biosciences, Drug Research Program, Faculty of Pharmacy, University of Helsinki, Helsinki, Finland. ✉e-mail: liisa.vilen@astrazeneca.com

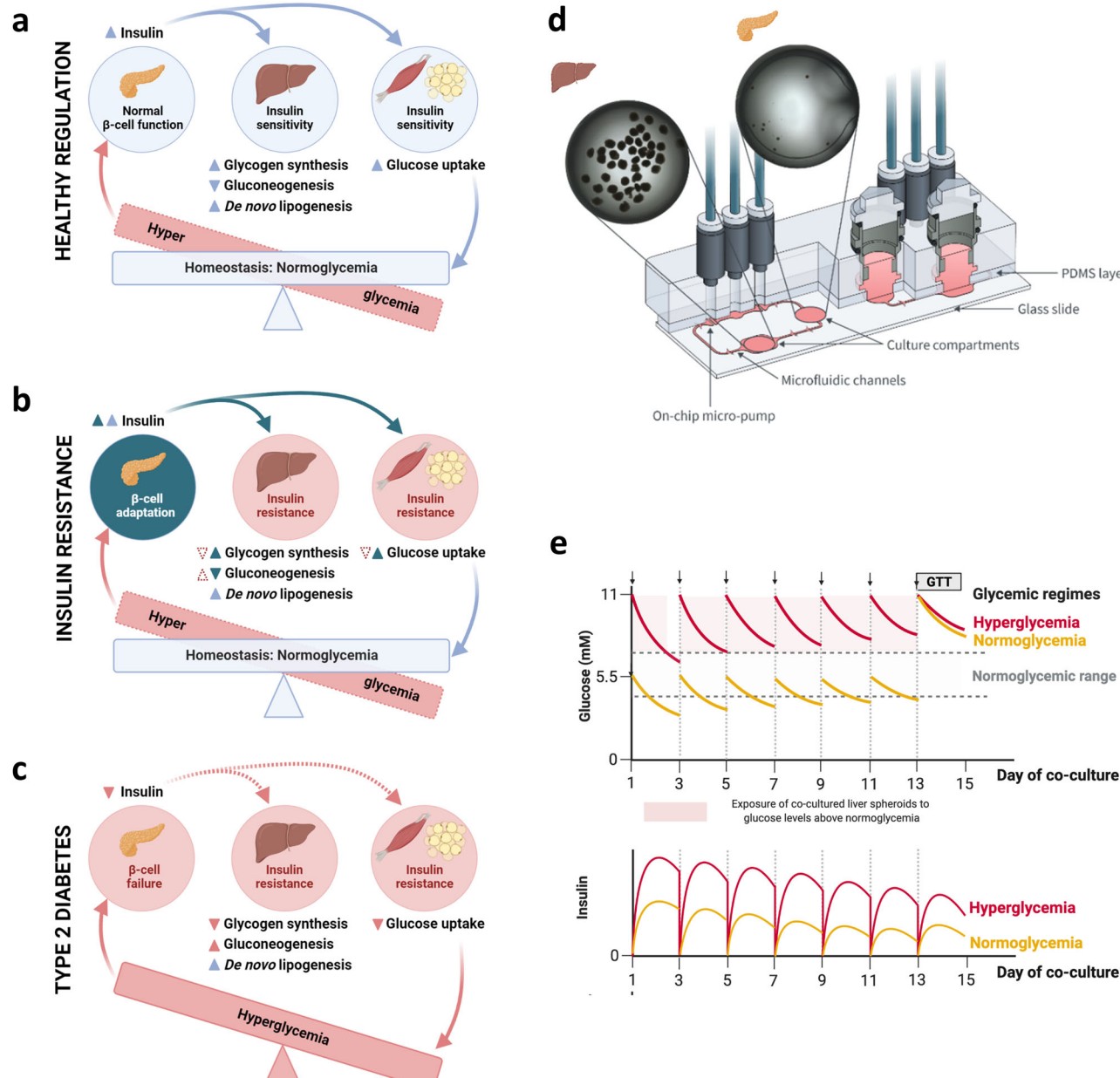

**Fig. 1 | Pancreas-liver MPS for investigation of diabetic glucose dysregulation.** **a** Healthy glucose regulation. Pancreatic islets prevent long-term hyperglycemia by secreting insulin which promotes glucose uptake and storage as well as de novo lipogenesis (▲) and inhibits glucose release (▼) in insulin-sensitive tissues including the liver, muscle, and adipose tissue. **b** Glucose regulation in the insulin-resistant state. Insulin resistance causes decreased glucose uptake and storage (▽) as well as increased glucose release (⟋) while insulin-stimulated lipogenesis remains unaffected (▲). Adaptive insulin secretion (▲▲) prevents hyperglycemia by normalizing glucose uptake and storage (▲) and inhibiting glucose release (▼). **c** Glucose regulation in type 2 diabetes. Long-term hyperglycemia develops due to a reduced insulin secretion (▼) which reduces glucose uptake and storage (▼) and increases glucose release (▲). **d** Schematic of the pancreas-liver co-culture in the Chip2 MPS. Each Chip2 has two separate circuits (left and right) which allows for the culture of two independent replicates on one chip. An on-chip

micropump generates a pulsatile flow through the microfluidic channels enabling cross-talk between the culture compartments. For the pancreas-liver MPS, 40 HepaRG/HHSteC liver spheroids are cultured in the outer culture compartment and 10 islets in the inner culture compartment of one circuit. Each circuit contains 605 μl of co-culture medium (5 μl in the microfluidic channels and 300 μl in each culture compartment). Medium is exchanged by replacing the total volume in each culture compartment. Brightfield images show both organ models on day 1 of co-culture in their respective compartment. **e** Graphical illustrations of representative, previously reported, glucose and insulin responses in the pancreas-liver MPS[14] visualize the development of glucose dysregulation over time. These responses were observed in seven independent pancreas-liver MPS studies corresponding to seven different donors of pancreatic islets. Arrows indicate medium exchange. GTT glucose tolerance test.

high doses of glucocorticoids, can induce hyperglycemia by influencing hepatic glucose production and insulin sensitivity[4] and even impair insulin secretion by pancreatic beta cells[5].

Due to diabetes mellitus being a multisystem disease, preclinical studies on its progression have traditionally relied on animal models. These models have historically been instrumental in discovering anti-

diabetic treatments and remain the primary experimental model for studying the complex pathophysiology and multi-organ interactions of diabetes[6]. However, animal models used in the diabetes research are genetically and physiologically different from humans limiting the translatability[7]. Additionally, these models are not well-suited for studying human-specific emerging drug classes such as oligonucleotides[8]

which directly target disease-causing genes and may have low cross-reaction to the corresponding genes in animals[9].

The recent advancements in microphysiological systems (MPS) such as organ-on-chip models have enabled human in vitro studies of physiological organ crosstalk, disease development, and pharmacological effects[10,11]. Since the pancreas and the liver are central organs in blood glucose regulation, we and others have shown that functional coupling of pancreatic and liver organ models on chip can recapitulate human-relevant pancreas-liver axis[12–14]. In these two-organ models, human islet microtissues (InSphero)[12,14] (Fig. 1d) or human induced pluripotent stem cell (hiPSC)-derived islet organoids[13] secrete insulin into the circulating co-culture medium. Secreted insulin was shown to stimulate glucose utilization in the liver model, composed of HepaRG hepatocytes and human hepatic stellate cells (HHSteC)[12,14] or hiPSC-derived liver organoids[13]. In response to the decreasing glucose concentration in the co-culture medium, the insulin secretion subsided demonstrating a physiological feedback loop between the liver and the islet compartment (Fig. 1e, hyperglycemia)[12,14].

In our pancreas-liver MPS, both glucose utilization and insulin response declined over time[12,14] (Fig. 1e, hyperglycemia), indicating its potential for studying glucose dysregulation and beta-cell dysfunction in vitro. However, using this disease model would require a healthy model as a control group. Initially, we suspected high glucose concentration (11 mM) as the primary driver for glucose dysregulation as hyperglycemia is known to induce insulin resistance in vitro[15] and in vivo[16]. Surprisingly, declining glucose utilization and reduced insulin response over the culture time were observed even in chips maintained in 5.5 mM glucose[14], mimicking normal blood glucose levels (Fig. 1e, normoglycemia). However, in the in vitro adjusted glucose tolerance (GTT) assay, the glucose reduction was driven by a significantly lower insulin secretion, suggesting a higher insulin sensitivity than in hyperglycemic cultures (Fig. 1e, day 13–15).

In this study, we extensively characterize the pancreas-liver MPS and investigate the factors influencing glucose dysregulation and beta-cell dysfunction. Due to the complex and dynamic nature of organ crosstalk, we combine the in vitro model with in silico modeling for hypothesis testing, data analysis, and informed decision-making. This approach allows us to study whether glucose dysregulation is driven solely by hyperglycemia, and we describe a physiological culture condition that rescues the glucose homeostasis and beta-cell function. Furthermore, we use RNA sequencing and proteomics analysis to profile soluble proteins, potentially affecting islet proliferation in the chip co-culture. Finally, we evaluate reproducibility, transferability, and the inter-donor variability of the pancreas-liver MPS across two laboratories and multiple islet donors.

## Results
### In silico-supported experimental design to resolve cues driving glucose dysregulation
To study the factors driving glucose dysregulation in the pancreas-liver MPS, we formed two hypotheses aimed at determining whether normoglycemia alone could improve the insulin sensitivity, as we previously thought. The first hypothesis (H1; Fig. 2a, left graph) assumes that insulin resistance stems from hyperglycemia, while the second hypothesis (H2; Fig. 2a, right graph) proposes that insulin resistance results from a combination of hyperglycemia and an additional diabetogenic factor. To study these hypotheses, we employed a computational hypothesis-testing approach (Fig. 2b; see "Methods" for details) using our recently described mathematical model of glucose and insulin interplay in the pancreas-liver MPS[14] (Supplementary Fig. 1a).

We conducted a 15-day chip study involving sequential experimental and modeling iterations to differentiate between the two hypotheses. Specifically, we exposed the hyperglycemic and normoglycemic chips to a defined insulin dose at the end of the co-culture, and then differentiated between H1 and H2 based on the glucose tolerance curves (Fig. 2c). First, we developed mathematical models for both H1 and H2 (Supplementary Fig. 1b) and then exposed the pancreas-liver MPS to either hyperglycemic or normoglycemic conditions for 13 days (Fig. 2c). Glucose and insulin

concentrations were recorded at the beginning (GTT assay on days 1–3) and in the middle (GTT assay on days 7–9) of the co-culture study, and these values were used to calibrate the mathematical models to account for donor-dependent variations in the insulin secretion. In this calibration step, both H1 and H2 provided acceptable agreement with the experimental data (Fig. 2d, e, left graphs), as confirmed by a statistical $\chi^2$ test (see "Methods" for details).

Next, we utilized the calibrated mathematical models to select an insulin dose that, when spiked to the co-culture medium, would yield distinct predictions for the glucose tolerance curves for hypotheses H1 and H2 (Supplementary Fig. 2). When performing the GTT assay with the suggested insulin dose (24 nM), we observed similar glucose response to 11 mM glucose load in both hyperglycemic and normoglycemic conditions (Fig. 2d, e, right graphs). Upon comparing the model predictions against the experimental data, we did not find a statistically acceptable agreement for H1 (Fig. 2d, right graphs), leading to the rejection of the hypothesis that insulin resistance was induced by hyperglycemia alone. In contrast, H2 aligned with the experimental data (Fig. 2e, right graphs) according to $\chi^2$ statistics. Therefore, we further investigated the hypothesis that glucose dysregulation was induced by a combination of hyperglycemia and an additional diabetogenic factor.

### Hydrocortisone and hyperglycemia impact islet and liver functionality on chip
The computational hypothesis testing approach indicated the involvement of an additional diabetogenic factor in the development of glucose dysregulation in the pancreas-liver MPS. Upon reviewing all the supplements used in the co-culture medium, we suspected that an unphysiological glucocorticoid concentration might be a contributing factor. Hydrocortisone (HCT) is used in the standard HepaRG culture medium[17], and therefore in our original chip co-culture medium[12], to ensure optimal differentiation and functionality of the liver model[18]. However, the supplemented concentration (50 μM)[17] exceeds the physiological plasma concentration (about 5.5–39 nM)[19] by several orders of magnitude. We hypothesized that this heightened HCT concentration might induce a diabetic phenotype in our pancreas-liver MPS, similar to that seen in patients suffering from glucocorticoid-induced or "steroid" diabetes. Conversely, reducing HCT to its physiological level was anticipated to rescue beta-cell function, insulin sensitivity, and overall glucose homeostasis by enhancing insulin secretion and preventing the induction of gluconeogenesis and steatosis in the liver model.

Glucocorticoids have also been shown to impair the insulin secretion of beta cells in vitro[5,20] and in patients susceptible to beta-cell dysfunction[5,20,21]. To identify an HCT concentration with minimal effects on glucose-stimulated insulin secretion (GSIS), we performed a dose-response analysis with islets cultured in our normoglycemic co-culture medium covering a broad concentration range (0.05 nM–50 μM; Fig. 3a). We observed an inhibitory effect on insulin secretion already at 50 nM HCT, whereas 5 nM HCT showed no difference compared to the untreated control. Next, we asked whether a lower, physiological concentration of HCT in the co-culture medium would, first, maintain liver functions and improve insulin secretion and, second, improve the overall glucose homeostasis. We set the tested low HCT concentration to 10 nM to be within the physiological plasma concentration range. To study all variables, we maintained pancreas-liver co-cultures for 2 weeks in four different medium conditions, using either high HCT (50 μM) or low HCT (10 nM) concentrations and either hyperglycemic (HG; 11 mM) or normoglycemic (NG, 5.5 mM) glucose concentrations (Fig. 3b). To account for donor-to-donor variability, we carried out three independent studies, each with a different islet donor in two laboratories (Studies 1 and 2 at TissUse, Study 3 at AstraZeneca; Supplementary Tables 1 and 2).

We monitored the HepaRG/HHSteC liver spheroids' functionality by following albumin secretion over time (Fig. 3c) and measured mRNA expression of key markers of liver health (Fig. 3d). We observed a stable albumin secretion at low HCT conditions, both in hyperglycemia and in

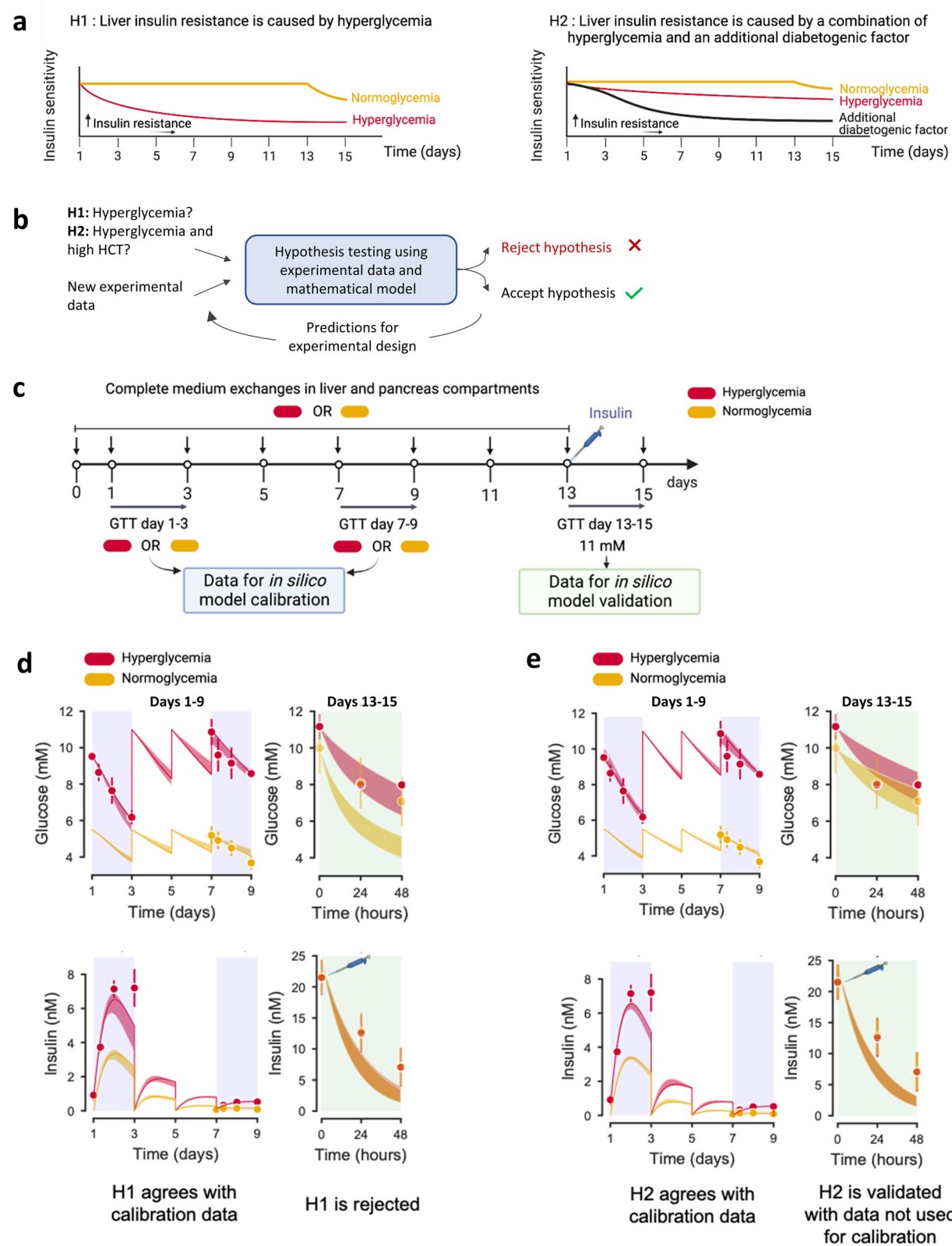

normoglycemia, while a high HCT concentration increased albumin secretion over time at both glucose concentrations (Fig. 3c). The lower HCT concentration had minimal effect on the mRNA of *HNF4A*, a key transcription factor of liver-specific genes, *ALB*, encoding the liver-secreted plasma protein albumin, *AHSG*, encoding the liver-secreted protein fetuin-A, and *MRP2*, encoding a canalicular multi-specific organic anion

transporter (Fig. 3d). In turn, the lower HCT concentration reduced the expression of *CYP3A4* mRNA, a major drug-metabolism enzyme, *CPS1* mRNA, an enzyme participating in urea production, and *ABCB11* encoding BSEP, the major bile-acid transporter. This, however, was not unexpected as glucocorticoids are known inducers of cytochrome P450 enzymes[22], the urea cycle (e.g., CPS)[23], and hepatic bile acid transport[24]. Furthermore, the

**Fig. 2 | In silico-supported experimental design to resolve cues driving glucose dysregulation. a**, **b** Schematic representation of the hypothesis testing framework to unravel the cause of insulin resistance in the pancreas-liver MPS. **a** Here, we considered two competing hypotheses (H1, H2) for the development of liver insulin resistance in the pancreas-liver MPS. Hypothesis H1 (left graph) assumes that insulin resistance is caused by hyperglycemia alone, while hypothesis H2 (right graph) assumes that insulin resistance is caused by hyperglycemia in combination with an additional diabetogenic factor. **b** Hypothesis testing is an iterative process, where mathematical models constructed from experimental data are used to test and reject hypotheses. **c** In silico guided experimental design to test the proposed hypotheses. The computational model was first calibrated for donor-dependent variations using data from a glucose tolerance test (GTT) on days 1–3 and 7–9. Next, the computational model was used to select an insulin dose that was added to the co-cultures on day 13. This experimental design would lead to different glucose tolerance curves on day 13–15 and allow differentiation between H1 and H2. Comparison between experimental data (dots) and model simulations (lines) in calibration phase (blue areas) and validation phase (green areas) for H1 (**d**) or H2 (**e**). The shaded areas in the simulated curves (red and yellow) represent the model uncertainty. Both H1 and H2 agree with calibration data for glucose (top) and insulin (bottom) but only H2 predicts glucose response during the validation step on days 13–15. Experimental data in (**d**, **e**) are shown as mean ± SEM, $n = 3$ individual co-culture replicates (circuits). Study was performed at AstraZeneca.

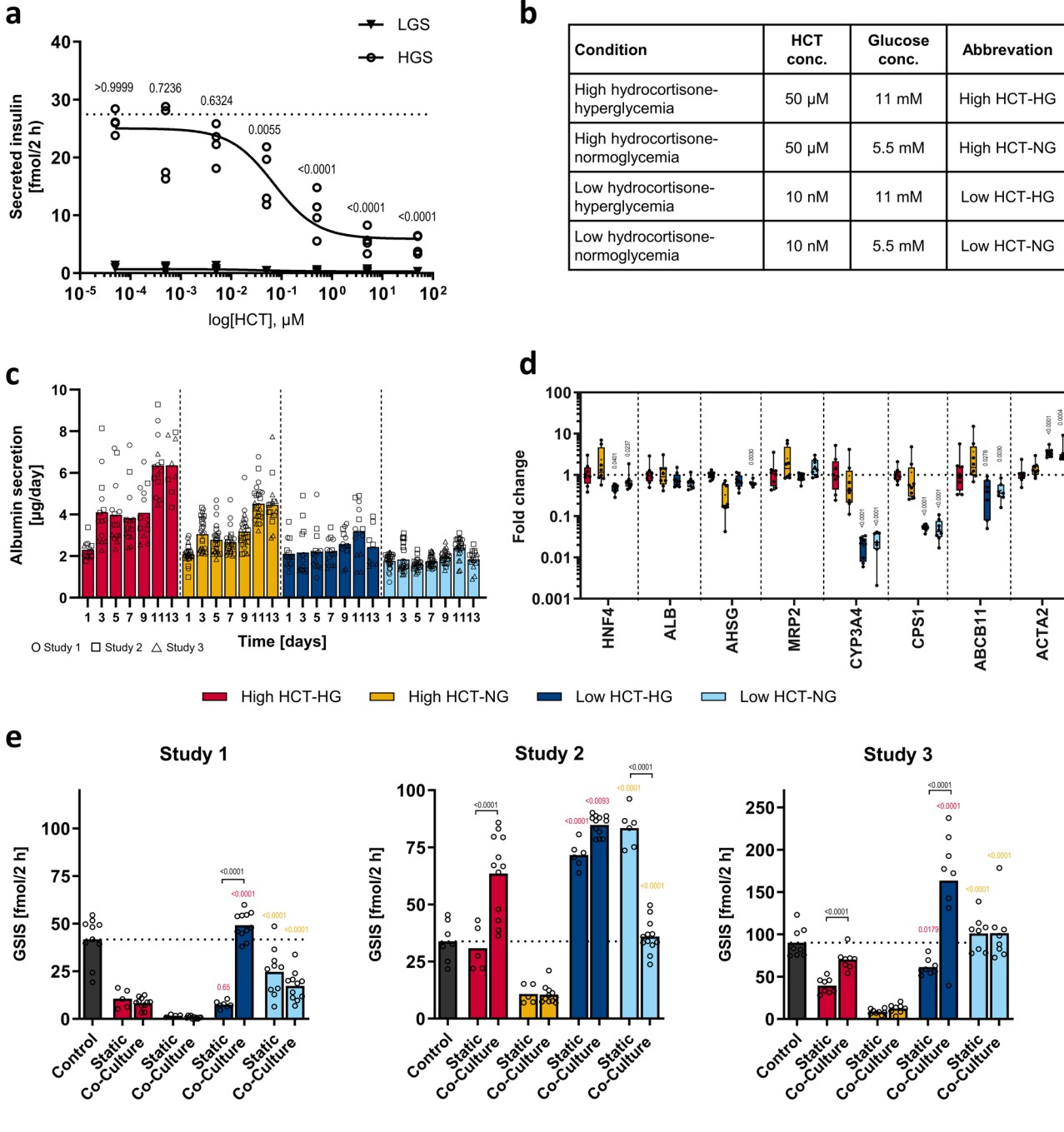

**Fig. 3 | Hydrocortisone and glucose concentrations impact liver and islet functionality. a** Hydrocortisone (HCT) concentration-dependent inhibition of glucose-stimulated insulin secretion (GSIS) of islets in static monoculture in normoglycemic co-culture medium. Symbols represent insulin amount secreted by individual islets ($n$ = 4) during a 2-h incubation in a low-glucose solution with 2.8 mM glucose (LGS, triangles) or a high-glucose solution with 16.8 mM glucose (HGS, circles). Data were fitted using nonlinear regression to retrieve a dose-response curve. IC50 was determined to be 70 nM (95% CI = 23, 240). Differences to the control (no HCT, represented by dotted line) were evaluated by one-way ANOVA using Bonferroni's multiple comparisons post hoc test. **b** Overview of the four different experimental conditions studied in the pancreas-liver MPS. **c** Liver spheroid functionality shown by albumin secretion over the chip co-culture time. Bars show mean and symbols represent individual co-culture replicates (circuits) from three independent studies ($n$ values summarized in Supplementary Table 1). **d** Gene expression of key liver markers in liver spheroids extracted at the end of co-culture. Data shown as fold changes to the high HCT-hyperglycemic condition in a box-whisker plot with median and min-max values. Symbols represent liver samples from individual co-culture replicates (circuits) from studies 1 and 2 performed at TissUse (Study 1:

$n$ = 4, Study 2: $n$ = 4). Differences between high and low HCT concentration for the same glucose concentration were evaluated by one-way ANOVA using Sidak's multiple comparisons post hoc test, $p$ values only shown for significantly different comparisons. **e** Pancreatic islet function (GSIS) after 15 days of static monoculture or chip-based co-culture in the four different co-culture media. Islets cultured for 15 days in static monoculture in culture medium provided by the manufacturer served as a control (black bar and dotted line). Islets were extracted from the MPS to perform the assay with individual islets. Bars show mean and symbols represent the insulin amount secreted by individual islets ($n$ values summarized in Supplementary Table 1) during a 2-h incubation in a high glucose solution (16.8 mM glucose). An individual donor was used for each study. Differences between selected pairs were evaluated by one-way ANOVA using Sidak's multiple comparisons post hoc test. Data were log-transformed for normality. Colored $p$ values show comparison between high and low HCT concentrations in hyperglycemia (red) or normoglycemia (yellow) in the same culture format (static or co-culture). Black $p$ values show significant differences between static monoculture and co-culture in the same medium. Studies 1 and 2 were performed at TissUse and Study 3 at AstraZeneca.

expression of *ACTA2* encoding alpha-smooth muscle actin was increased, suggesting that HHSteCs proliferate when the HCT concentration is reduced. This was also expected, as glucocorticoids are known for their anti-fibrotic effects[25]. In general, albumin secretion and the expression of liver-specific genes were preserved at the low HCT concentration, but some metabolic functions might be reduced compared to co-cultures maintained at high HCT concentrations.

Next, we studied the effect of the four different media on islet functionality by analyzing the GSIS of islets extracted from the chips after the dynamic co-culture as well as of islets monocultured statically (Fig. 3e). As demonstrated in three independent studies with individual islet donors, media with low HCT concentration improved the GSIS as compared to high HCT concentration both in hyperglycemia and in normoglycemia (Fig. 3e; Basal insulin secretion and stimulation index reported in Supplementary Fig. 3). In studies 2 and 3, the islet functionality in the most physiological condition, i.e. normoglycemic with low HCT, was similar or better as compared to the control culture where the islets were maintained in the organ-specific medium provided by the islet manufacturer. In addition, we observed that co-cultures in hyperglycemic media significantly increased the insulin secretion compared to static monocultures (Study 1: Low HCT-HG, Study 2: High HCT-HG, Study 3: High HCT-HG and Low HCT-HG).

### Physiological culture condition rescues glucose regulation in the pancreas-liver MPS

To further evaluate whether a lower HCT concentration or normoglycemic glucose concentration, or these together, would lead to improved glucose regulation, beta-cell function, and insulin sensitivity during the pancreas-liver co-culture, we performed a GTT assay on days 1–3 (only hyperglycemic conditions) and on days 13–15 (all four conditions). To determine if the measured glucose and insulin responses could be explained by our hypothesis H2, we applied the in silico modeling approach. First, we calibrated the computational model corresponding to hypothesis H2 using the experimental glucose and insulin concentrations from co-cultures exposed to high HCT (Fig. 4a). Then, we used the calibrated model to predict the expected insulin and glucose responses assuming that the lower HCT concentration would not affect the insulin sensitivity, and the insulin secretion capacity would be maintained. By comparing these predictions to our experimental data, we found that the computational model can explain the measured responses indicating that low HCT concentration can indeed maintain the insulin sensitivity and beta-cell function in the pancreas-liver MPS (Fig. 4b).

Total glucose utilization and insulin secretion were also analyzed by calculating the areas under the glucose and insulin concentration curves (AUCs). Low HCT conditions maintain glucose tolerance and preserve islet functionality over time, seen as relatively stable AUCs for glucose and

insulin, while in the high HCT the AUCs for glucose increases over time and the AUCs for insulin drastically drop over time (Fig. 4c). Confirming these findings, we saw similar responses in a repeated pancreas-liver MPS study with the difference that the glucose tolerance was only maintained in the low HCT-normoglycemic condition (Supplementary Fig. 4).

Altogether, we show that a "healthy" pancreas-liver co-culture with stable liver function, beta-cell function, and glucose tolerance is achieved in a condition with low HCT concentration and normoglycemic glucose level. In contrast, a "diseased" co-culture representing impaired glucose tolerance accompanied by beta-cell dysfunction can be generated by using a high HCT-hyperglycemic medium. Therefore, we further focused on these two co-culture conditions and data on the two intermediary conditions can be found in Supplementary Figs. 5–8.

### Hepatic phenotype reflects glucocorticoid-induced diabetes

In patients with glucocorticoid-induced diabetes, hepatic insulin resistance is one factor contributing to dysbalanced glucose regulation and hyperglycemia[21]. Glucocorticoids increase endogenous glucose production by inducing the transcription of genes encoding gluconeogenic enzymes (e.g. glucose-6-phosphatase)[26,27]. Moreover, glucocorticoids induce glycogen synthesis[28] which increases the liver's capacity to produce glucose. Furthermore, chronic elevation of glucocorticoid concentration has been linked to the development of a steatotic 'fatty' liver by increasing the gene transcription of several enzymes involved in de novo lipogenesis, including the fatty acid synthase[26]. Excess fatty acids are partly converted to ketone bodies, leading to elevated ketone levels in plasma[26,29].

To analyze if our liver model reflects the glucocorticoid-induced diabetic phenotype, we first looked at gene expression profiles of enzymes involved in glucose metabolism (Fig. 5a), ketogenesis (Fig. 5b), and lipid metabolism (Fig. 5c) in the co-cultured HepaRG/HHSteC spheroids. The diseased condition induced gene expression of glycogen synthase (*GYS2*) involved in glycogen synthesis, glucose-6-phosphatase (*G6PC*) involved in gluconeogenesis, HMG-CoA lyase (*HMGCL*) involved in ketogenesis, and fatty acid synthase (*FASN*) involved in de novo lipogenesis. Next, we confirmed these findings by performing separate analyses to evaluate glycogen storage, ketone-body production, and lipid metabolism in the co-cultured HepaRG/HHSteC liver spheroids. Liver spheroids in the diseased co-cultures exhibited higher amounts of glycogen stores as shown by periodic acid-Schiff (PAS) staining (Fig. 5d), secreted 2.6-fold more 3-hydroxybutyrate (Fig. 5e), a diagnostic measure of diabetic ketoacidosis[30], and accumulated more intracellular lipids as visualized by Nile Red staining (Fig. 5f) when comparing to spheroids in the healthy condition. These data indicate that the diseased liver model reflects similar pathological alterations as seen in patients suffering from steroid diabetes.

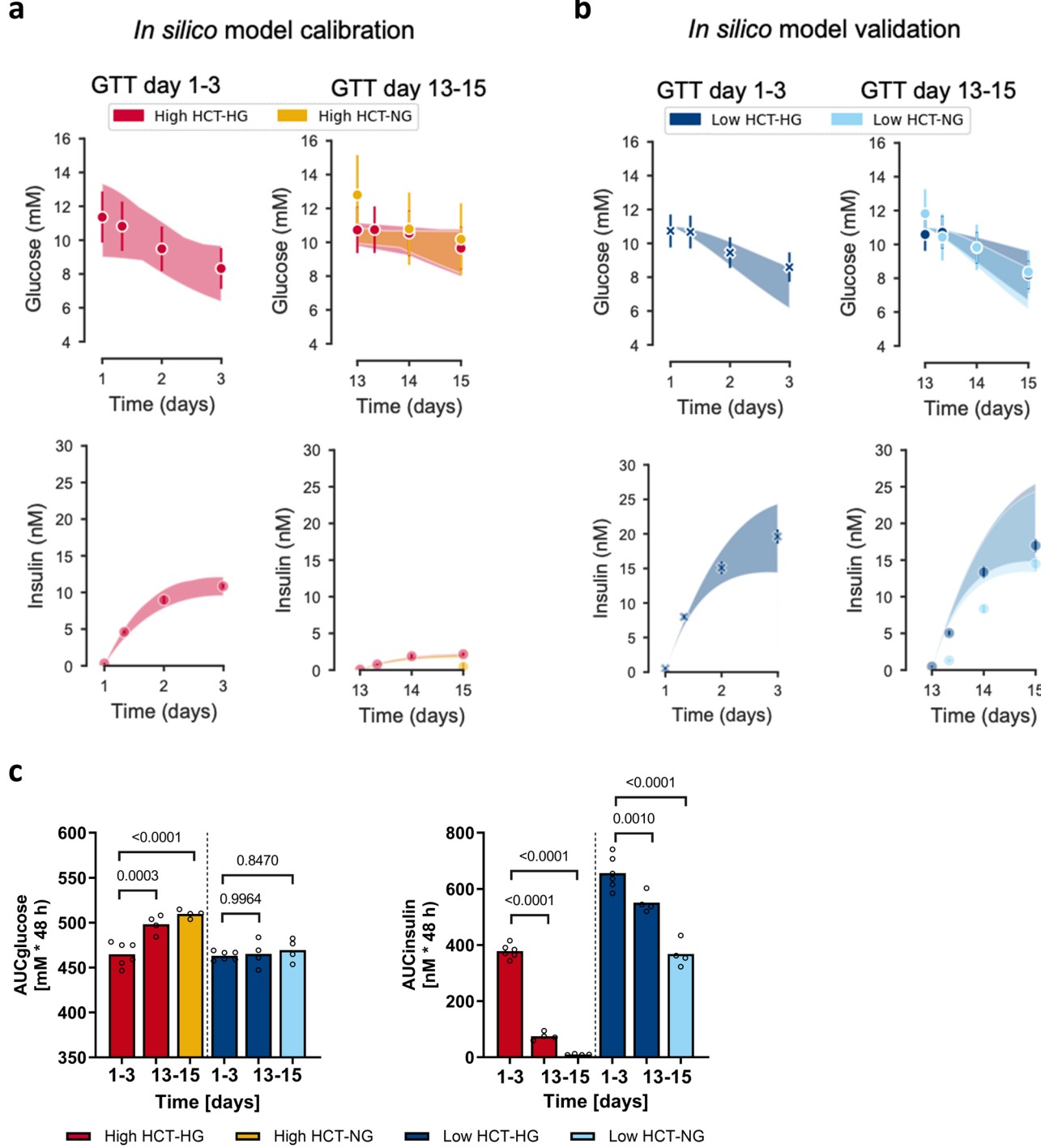

## Combined omics analysis to examine factors promoting islet proliferation on chip

Individuals with insulin resistance do not necessarily develop glucose dys-regulation and diabetes as beta cells can compensate for the increased insulin demand by either increasing in number (proliferation or transdifferentiation) or enhancing their secretory output, or both[31]. Previously, several studies have demonstrated that organs, including the liver, secrete proteins into the bloodstream which stimulate insulin secretion and proliferation of islets[32]. Since we had observed that especially the hyperglycemic chip co-cultures improve the insulin secretion as compared to the static monocultures (Fig. 3e), we asked if this could be explained by an increased islet cell number. To study this, we developed a cell proliferation assay using 5-ethynyl-2'-

deoxyuridine (EdU) incorporation, automated high-throughput confocal microscope imaging, and automated image analysis (Supplementary Fig. 9). Surprisingly, when the islets were cultured in the disease condition, pro-liferation did not differ between the chip co-culture and the static mono-culture (Fig. 6a). This suggests that other beta-cell adaptation mechanisms than increased cell mass contribute to the improved insulin secretion capacity seen in the chip co-cultures (Fig. 3e). Instead, in the healthy condition, pro-liferation was significantly increased in the co-cultured islets as compared to the monocultured islets (Fig. 6a). Additionally, the chip co-cultured islets were more proliferative in the healthy condition than in the disease condition.

To identify potential liver-secreted proteins influencing the islet pro-liferation in the co-cultures, we performed exploratory transcriptome and

**Fig. 4 | Physiological hydrocortisone level maintains glucose tolerance in the pancreas-liver MPS. a, b** MPS combined with in silico modeling to study the effect of HCT on insulin sensitivity. The mathematical model is used to predict glucose and insulin responses at physiological HCT concentrations that agree with experimental measurements, indicating that low HCT can maintain insulin sensitivity and beta-cell function. **a** The mathematical model was calibrated using experimental data measured in pancreas-liver MPS exposed to high HCT concentration. Experimental measurements of glucose (dots, top row) and insulin (dots, bottom row) were acquired during GTT assays on days 1–3 and 13–15 in hyperglycemic high-HCT (red) and normoglycemic high-HCT (yellow) conditions. **b** The calibrated model was then used to predict glucose (top row) and insulin responses (bottom row) at physiological HCT concentration. These predictions were compared to the corresponding experimental GTT measurements on day 1–3 (left) and 13–15 (right) from hyperglycemic low-HCT condition (dark blue) and normoglycemic low-HCT (light blue). Both the model predictions and the experimental measurements show that

glucose tolerance is maintained at day 13–15 (top row, right) compared to day 1–3 (top row, left), and that insulin levels at the end of the co-culture are also similar at day 13–15 (bottom row, right) compared to day 1–3 (bottom row, left) for both hyperglycemic low-HCT (dark blue) and normoglycemic low-HCT (light blue) conditions. Data points from GTT measurements of glucose and insulin on day 1–3, which are depicted with an X, were used for baseline correction of insulin sensitivity and insulin secretion capacity (see "Methods" for details). The shaded areas in (**a, b**) represent the model uncertainty. Experimental data shown as mean ± SEM from Study 1 performed at TissUse (day 1–3: $n = 6$ individual co-culture replicates, day 13–15: $n = 4$ individual co-culture replicates). **c** Area under the curve (AUC) for glucose (left) and insulin (right). Bars show mean and symbols represent individual co-culture replicates (circuits) from Study 1 performed at TissUse (day 1–3: $n = 6$, day 13–15: $n = 4$). Differences between selected pairs of conditions (day 13–15 compared to day 1–3) were evaluated by one-way ANOVA using Sidak's multiple comparisons post hoc test.

proteome analyses of liver spheroid lysates and supernatants from hyperglycemic and normoglycemic co-culture conditions after 2-week chip co-culture. In a combined RNA sequencing and proteomics analysis, IL-1R2 was the most upregulated protein in the hyperglycemic condition (Fig. 6b). IL-1R2 exists both as soluble and membrane-bound decoy receptor, a competitive inhibitor preventing IL-1beta signaling[33]. IL-1beta is an inflammatory cytokine associated with beta-cell dysfunction and reduced proliferation capacity[34], thus a potential target for diabetes therapies. It has been shown before in in vitro studies, animal models, and clinical trials, that inhibition of interleukin-1 receptor (IL-1R) and therefore a prevention of IL-1beta signaling enhanced beta-cell survival and function[35–38]. Therefore, we hypothesized that IL-1R2 could as well enhance beta-cell proliferation and function by reducing the detrimental free IL-1beta concentration. To test this, we first quantified soluble IL-1R2 in the chip co-culture supernatant over time and confirmed a significant upregulation of IL-1R2 secretion in the disease condition (Fig. 6c). To increase confidence that IL-1R2 is primarily produced by the liver model, we analyzed the soluble IL-1R2 secretion in the supernatant from static monocultured islets and found no secretion (Supplementary Fig. 10a). Next, we treated islets in static monoculture with 30 ng/ml or 0.3 ng/ml of recombinant human IL-1R2 mimicking the measured levels in the diseased and healthy condition, respectively. We cultured the islets in the medium provided by the islet supplier since the chip co-culture media induce proliferation (Supplementary Fig. 7), and we wanted to decouple the effect of IL-1R2 from any medium-induced proliferation. Compared to untreated control, we observed a 4.9-fold increase in proliferation measured as a proportion of EdU-positive cells in islets treated with 0.3 ng/ml of IL-1R2 (Fig. 6d). In contrast, 30 ng/ml of IL-1R2 did not affect proliferation. These results reflect our findings from the pancreas-liver MPS (Fig. 6a) where we observed higher proliferation in the islets in the healthy condition (low IL-1R2 levels) compared to islets in the diseased condition (high IL-1R2 levels). In line with an earlier observation that proliferating beta cells have an impaired insulin response[39], we observed reduced glucose-stimulated insulin secretion at low IL-1R2 concentration (Supplementary Fig. 10b).

### Reproducibility of the pancreas-liver MPS

MPS studies are generally complex with multiple confounding factors that can cause variability during and between the studies[40]. Here, we examined technical sources of variability and the reproducibility of key biological effects in the pancreas-liver MPS, focusing on on-chip readouts relevant for future decision-making.

Using a statistical mixed model approach, we decomposed the variance according to technical factors present: between the laboratories, studies (performed in the same laboratory), and circuits. For each readout included in this analysis, we explored a set of mixed models to find the most appropriate set of random effects to represent the data. This was done by evaluating different metrics for model fits, as well as exploring how variance is allocated across technical factors. We also explored the proportion of

variance allocated to residuals—this is observation-to-observation variability that is not attributable to any other factor. Then, we compared the optimal statistical models across the readouts to identify how the variance decomposition varies across different readouts.

Our statistical model, incorporating nested random effects for laboratory, study, and circuit, revealed that most of the variability is allocated to residuals, rather than to technical factors. In the liver-specific readouts, the residuals account for 41.4 and 53.2% of the total variation in albumin secretion (Fig. 7a) and ketone-body production (Fig. 7b), respectively. Study-to-study variability contributes minimally to the overall variance (0.206% for albumin and 0% for ketone bodies). Since the study-to-study variability is negligible compared to the other factors, we conclude that it is not necessary to incorporate it in the statistical model. Similarly, residuals explain most of the variation for IL-1R2 secretion (Supplementary Fig. 11a). Lab-to-lab variation accounted for 13.0–29.6% of the total variance (Fig. 7a, b and Supplementary Fig. 11a), similar to circuit-to-circuit variation accounting for 10.1–33.8% of the total variance. In the pancreas-related readouts, donor-dependent variability is evident, as expected. The study-to-study variation accounts for 86.2% of the total variation in the GTT insulin response but only for 10.6% in the glucose utilization (Supplementary Fig. 11b, c). The lab-to-lab variability is not considered in the statistical analysis of the GTT insulin and glucose readouts as these studies were done in one laboratory.

Next, we examined the reproducibility of key biological effects across the studies and laboratories, focusing on the effect of diseased versus healthy condition. To estimate these effects, we applied mixed models to each study independently, and extracted both the effects and their standard errors. For albumin, standard errors of estimates were very similar across studies, and the estimated effect sizes were similar within laboratories (Fig. 7c). For the ketone bodies, the standard errors were again similar although there was more variability in the estimated effects and especially no effect in Study 3 (Fig. 7d). Like albumin, IL-1R2 production in the pancreas-liver MPS is very reproducible between the laboratories (Supplementary Fig. 11d). Also, for insulin and glucose readouts, the estimates and standard errors were of comparable magnitude across studies (Supplementary Fig. 11e, f).

Finally, we assessed intra-study and inter-study reproducibility using a recently published statistical methodology[41] based on intra-class correlation coefficient (ICC) and the maximum coefficient of variation (CV). In this analysis, the intra-study reproducibility of albumin, ketone bodies, and IL-1R2 was classified as acceptable to excellent (Table 1). Additionally, the reproducibility of glucose and insulin measurements during the GTT assay, the primary context of use for the model, was classified as excellent. In the inter-study assessment, both healthy and diseased conditions showed acceptable or excellent reproducibility for all readouts, with the exception of ketone bodies, which was classified as poor in the healthy condition (Table 2). Results for all on-chip readouts from each individual circuit in all studies are presented in Supplementary Fig. 12. Overall, the statistical

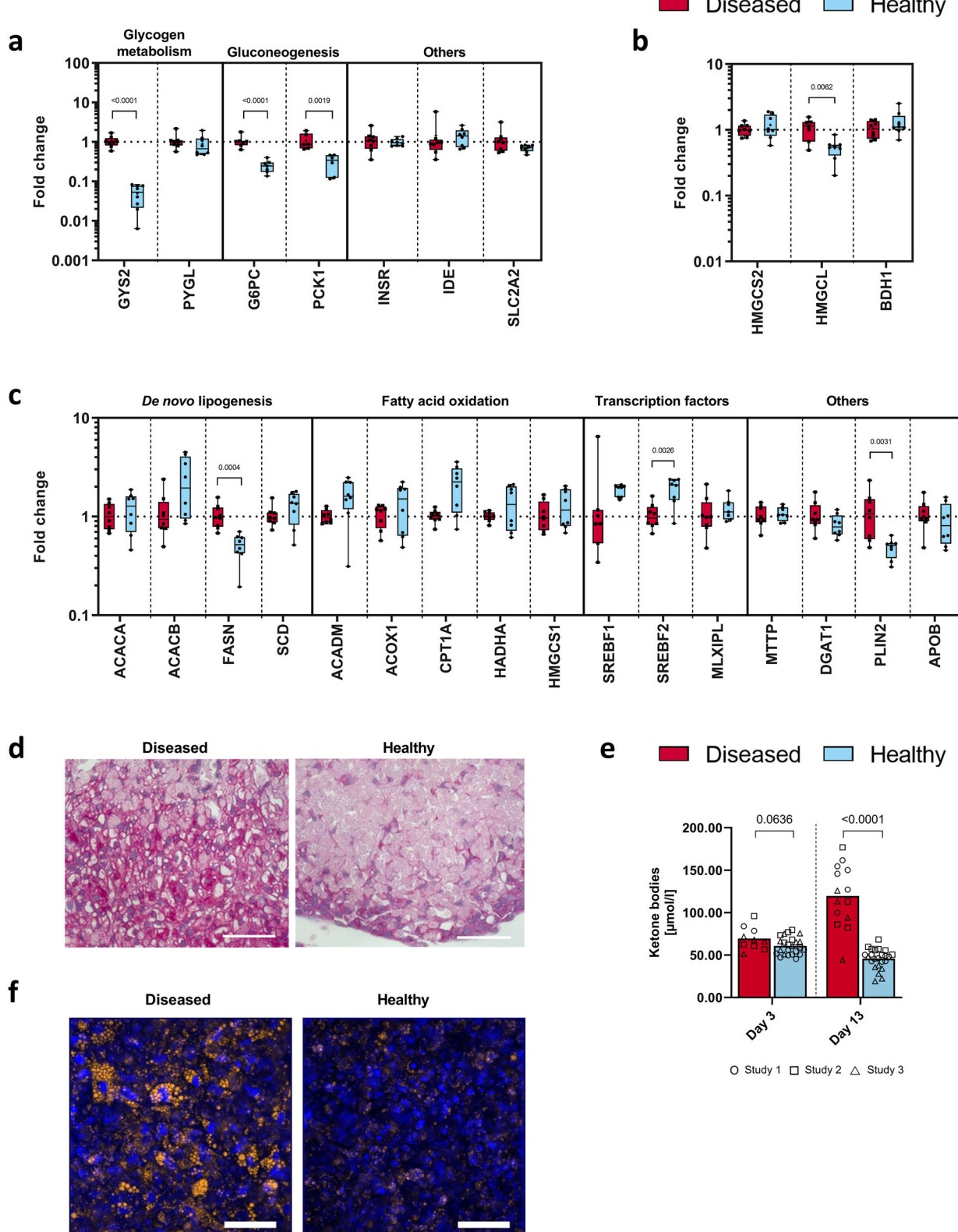

analyses demonstrated that we can draw reproducible conclusions across the on-chip readouts.

## Discussion

In this study, we describe a healthy pancreas-liver MPS with stable glucose homeostasis and a diseased MPS replicating hallmark features of glucocorticoid-induced diabetes. We set out to investigate the cues driving glucose dysregulation in the pancreas-liver MPS and demonstrated that a combination of hyperglycemia and high cortisone concentration leads to decreased glucose tolerance and islet dysfunction on chip. Subsequently, we showed that glucose homeostasis and islet functions can be rescued with physiological culture conditions. Defined physiological, healthy condition

**Fig. 5 | Hepatic phenotype reflects glucocorticoid-induced diabetes.** Gene expression of enzymes involved in hepatic glucose metabolism (**a**), ketone-body synthesis (**b**), and lipid metabolism (**c**) in liver spheroids extracted at the end of co-culture. Data shown as fold change between diseased (11 mM glucose, 50 μM HCT) and healthy (5.5 mM glucose, 10 nM HCT) conditions in a box-whisker plot with median and min-max values. Symbols represent liver samples from individual co-culture replicates (circuits) from studies 1 and 2 performed at TissUse (Study 1: $n = 4$, Study 2: $n = 4$). Differences between the diseased and the healthy condition were evaluated by multiple $t$-tests using the Holm–Sidak method for multiple comparisons without assuming a consistent standard deviation, $p$ values only shown for significantly different comparisons. **d** Glycogen storage visualized by periodic acid-Schiff (PAS) staining. Scale bar, 50 μm. **e** Ketone-body synthesis represented by 3-hydroxybutyrate concentration in the co-culture supernatants. Bars show mean and symbols represent co-culture replicates from three independent studies ($n$ values summarized in Supplementary Table 1). Studies 1 and 2 were performed at TissUse and Study 3 at AstraZeneca. Differences between the diseased and the healthy condition on day 3 or day 13 were evaluated by multiple $t$-tests using the Holm–Sidak method for multiple comparisons without assuming a consistent standard deviation. **f** Intracellular lipid droplets visualized by Nile Red staining (amber color). Blue denotes DAPI-stained nuclei. Scale bar, 50 μm.

—5.5 mM glucose and 10 nM HCT—enables studies on disease progression and mechanisms and this type of control group could even be used for examining metabolic disorders or development of insulin resistance caused by other disease mediators, such as free fatty acids, fructose, and cytokines[42]. Our findings establish the pancreas-liver MPS as a valuable human cell-based model for studying disease mechanisms of glucose dysregulation and a tool for drug discovery and development.

Previously, we suspected that hyperglycemia is the primary driver of the glucose dysregulation in the pancreas-liver MPS[12,14], given its known role in inducing insulin resistance[15,16]. To test this, we cultured chips in normoglycemia and observed decreased glucose tolerance during the GTT assay[14], suggesting the involvement of additional factors. In this study, we employed our recently developed mechanistic mathematical model[14] to guide hypothesis testing, support experimental design, and interpret data. Through a combined experimental and in silico approach, we confirmed that hyperglycemia alone does not explain the impaired glucose homeostasis in the pancreas-liver MPS. We then investigated the influence of unphysiological HCT concentration on glucose homeostasis, insulin sensitivity and beta-cell function. Glucocorticoids are known inducers of whole-body insulin resistance[5,20,21] and their intake can lead to the development of a condition called glucocorticoid-induced or steroid diabetes. We studied the effect of glucose and cortisone by maintaining the chips in four different medium conditions: (1) hyperglycemia and high HCT, (2) normoglycemia and high HCT, (3) hyperglycemia and low HCT, and (4) normoglycemia and low HCT. We utilized the in silico model to predict glucose and insulin responses and demonstrated that HCT is a key factor inducing glucose dysregulation and beta-cell failure in the pancreas-liver MPS. This experimental-computational hybrid approach is important for the accurate interpretation of multi-organ MPS data, particularly due to the complexity of cross-organ feedback loops, which are challenging to unravel by pure reasoning. Importantly, computational modeling enables in vitro–to–in vivo translation. We recently showed that pancreas-liver MPS results can be translated to humans by using mechanistic mathematical modeling, even when some MPS characteristics do not mirror human physiology, such as cell-to-liquid ratio and the flow rate, as these can be adjusted in the mathematical models[14].

The mathematical model indicated that the normoglycemic low-HCT condition maintains insulin sensitivity, while high HCT contributes considerably to the insulin resistance. Glucocorticoids increase hepatic glucose production via gluconeogenesis[43] and promote hepatic lipid accumulation[26], suspected to induce insulin resistance[44]. In our diseased co-culture condition, liver spheroids exhibited higher expression of genes involved in glycogen synthesis, gluconeogenesis, ketogenesis, and fatty acid synthesis, along with increased lipid storage, glycogen accumulation, and ketone-body secretion, reflecting some of the pathological alterations seen in patients suffering from glucocorticoid-induced diabetes. We also observed that HCT increases the albumin secretion both at normoglycemia and hyperglycemia. This might be an initial sign of developing insulin resistance as patients with elevated serum albumin concentrations have an increased risk of developing type 2 diabetes[45]. However, albumin secretion was not normalized to total protein amount so the effect of cell proliferation cannot be excluded. Overall, the functional and phenotypic characterization of the diseased liver spheroids parallels pathological alterations seen in steroid diabetes,

suggesting the development of glucocorticoid-induced insulin resistance. Further functional readouts could include a measurement of endogenous glucose production to confirm insulin resistance development in the liver spheroids.

Islets from different individuals vary in their beta-cell adaptation abilities[46]. Our GSIS assay showed that the hyperglycemic chip co-culture improved the insulin secretion compared to the static monocultures (Fig. 3e). This aligns with the enhanced beta-cell function observed in healthy and prediabetic individuals in response to rising blood glucose levels[5,19–21]. Thus, our co-culture model might reflect this typical beta-cell adaptation mechanism. Comparing three pancreas-liver MPS studies revealed high variability in the donors' ability to increase beta-cell function in response to hyperglycemia and, hence, adapt to the suspected development of insulin resistance in our pancreas-liver MPS. This reflects the typical donor-dependent variation in the beta-cell adaptation[46] and varying susceptibility to beta-cell failure due to diabetogenic factors such as glucocorticoids[20].

In preclinical diabetes studies, animal models are essential for investigating multi-scale mechanisms spanning different timescales and biological levels[47]. Therefore, they are well suited to study key mechanisms related to metabolic disorders such as body-fat distribution, systemic glucose metabolism, or brain control over metabolic fluxes[6]. However, the animal models differ genetically and physiologically from humans[7]. For example, rodent islets are known to have higher beta-cell adaptation capacity via proliferation compared to humans[48,49] and, thus, may not be an ideal model for finding human-relevant targets. In this study, we observed pronounced islet proliferation in the chip co-culture. We employed RNASeq and proteomics analysis to identify proteins that could account for the increased proliferation during the dynamic chip co-culture. In this exploratory work, we showed that there is 100-fold higher secretion of IL-1R2 in the diseased chips compared to healthy ones. Treating static monocultured islets with IL-1R2 showed that only the low concentration (reflecting the measured levels in healthy chips) stimulated proliferation, aligning with reports of low IL-1beta concentration benefiting islet functionality[50,51]. Therefore, it may be possible that the low IL-1R2 concentration in the healthy condition might have reduced the IL-1beta concentration to a beneficial range while the high IL-1R2 concentrations resulted in ineffectively low IL-1beta concentrations. Our findings suggest that chip-born IL-1R2 may impact the islet proliferation in the pancreas-liver co-culture but more studies are needed to better understand IL-1R2-mediated islet proliferation. Nevertheless, our work on chip-born factors that modulate the islet proliferation demonstrates, as a proof-of-concept, that multi-organ MPS can be used to find and study new targets and therapeutic proteins. Finally, it is important to note that all studied co-culture media, including the healthy medium, induced proliferation compared to the supplier's islet medium. Therefore, the islet phenotype might represent an initial pre-diabetic stage where islets compensate by increasing mass[31]. This surge in basal proliferation may also be influenced by fetal bovine serum, commonly used to support cell growth. Future work will investigate serum-free medium to achieve a pancreas model that fully reflects a healthy state.

In this work, we conducted three independent pancreas-liver MPS studies and used a mixed model statistical analysis to decompose the

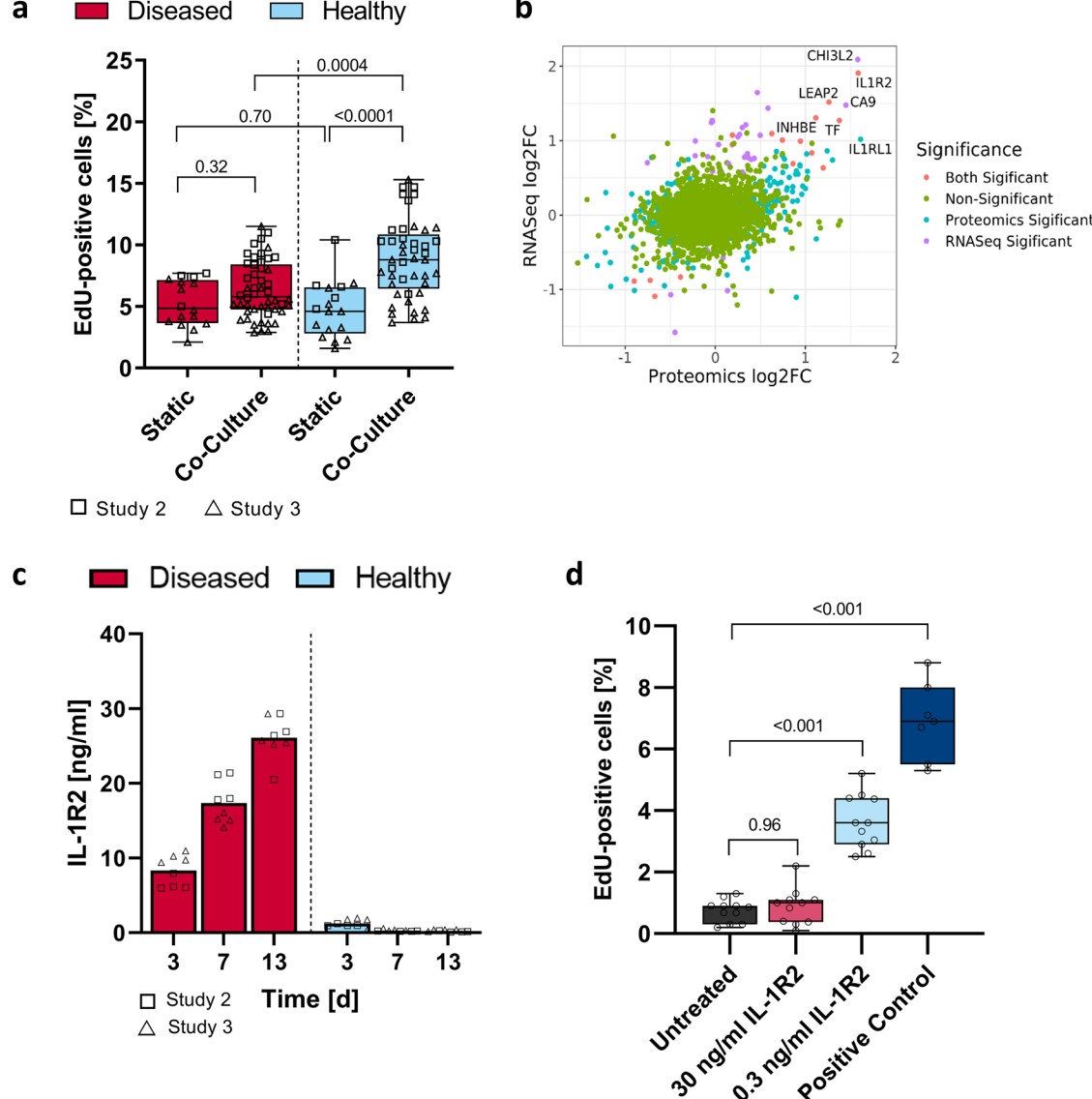

**Fig. 6 | Assessment of soluble factors promoting islet proliferation on chip.**
**a** Proliferation of islets in static monocultures and in chip co-culture with liver spheroids. In both culture systems, islets were maintained in the diseased (11 mM glucose, 50 μM HCT) and in the healthy (5.5 mM glucose, 10 nM HCT) condition and extracted for the proliferation assay at the end of the culture. Data shown as percentage of EdU-positive cells in a box-whisker plot with median and min-max values. Symbols represent individual islets from two independent co-culture studies (*n* values summarized in Table S1). Study 2 was performed at TissUse and Study 3 at AstraZeneca. Differences between selected pairs of conditions were evaluated by one-way ANOVA using Sidak's multiple comparisons post hoc test. **b** Multi-omics analysis on the effect of hyperglycemia vs. normoglycemia on liver-secreted proteins. Data from RNA-Seq (HepaRG/HHSteC liver spheroids) and proteomics (co-culture supernatants) were merged at the gene level. Point color indicate significance of change (FDR < 0.05 and *p* < 0.05 for RNA-Seq and proteomics data, respectively). Gene names are marked for genes with RNA-Seq and proteomic log2 fold-changes >1. Transcriptomics data are from three independent studies and proteomics data from four independent studies (see Supplementary Tables 1 and 2 for details). Two

studies were performed at TissUse (matching donors in RNA-Seq and proteomics analysis) and two studies were performed at AstraZeneca (one combined RNA-Seq and proteomics analysis with a matching donor and one additional donor in proteomics analysis). **c** IL-1R2 concentration in the chip-based co-cultures over time. Bars show mean and symbols represent individual co-culture replicates (circuits) from Studies 2 and 3 (Study 2: *n* = 4, Study 3: *n* = 4). Study 2 was performed at TissUse and Study 3 at AstraZeneca. **d** IL-1R2 treatment of islet monocultures. IL-1R2 stimulates cell proliferation at low dose (0.3 ng/ml) but not at the high dose (30 ng/ml) in islets monocultured in static condition in culture medium provided by the islet manufacturer. The islets were exposed to IL-1R2 for 16 days, and for the last 5 days of culture, medium was supplemented with 10 μM EdU for proliferation analysis. Positive control: hyperglycemic low-HCT co-culture medium. Data shown as percentage of EdU-positive cells in a box-whisker plot with median and min-max values. Symbols represent individual islets (*n* = 11 for untreated, 30 ng/ml IL-1R2 and 0.3 ng/ml IL-1R2, *n* = 7 for positive control). Differences to the untreated control were evaluated by one-way ANOVA using Dunnett's multiple comparisons post hoc test. Study was performed at AstraZeneca.

variability between the laboratories, studies, and circuits. Variability is often considered undesirable when designing in vitro systems, but trying to remove it can lead to the standardization fallacy[52] and less reproducible results. Working with that variability, rather than attempting to remove it, yields results that better generalize to the target patient population. We found that the lab-to-lab and circuit-to-circuit variability contributed to the

total variance at a similar magnitude for the analyzed on-chip readouts, including albumin, ketone bodies, and IL-1R2 production. Moreover, consistent standard errors across laboratories confirmed the successful transfer of the model from one laboratory to another without a substantial increase in technical variability. For the future multi-organ MPS studies, we intend to use the randomized block design proposed for the HUMIMIC

**Fig. 7 | Evaluation of technical sources of variation and reproducibility.** We analyzed data from on-chip readouts across all studies using a mixed model, which is a statistical model that can account for different sources of variability. By using the output from this model, we can understand better how technical factors affect the endpoints of interest and can show that even in the presence of this technical noise, we can still pick up biological effects of interest with sufficient precision. **a, b** The proportion (%) of total variance associated with each of the technical factors (circuit, study, laboratory) and the proportion of residual variance (that is, the variance that cannot be attributed to any of the other factors). Proportions are plotted respectively for the endpoints albumin (**a**) and ketone bodies (**b**). **c, d** We viewed the difference between the diseased and healthy conditions on day 13 as a benchmark biological difference of interest. Then, we used the mixed model to both estimate that difference, and also quantify how precise our estimate is. For each study, the difference and its associated standard error are shown for albumin (**c**) and ketone bodies (**d**). Note that in this case, the statistical model was applied to each study independently. Studies 1 and 2 were performed at TissUse and Study 3 at AstraZeneca.

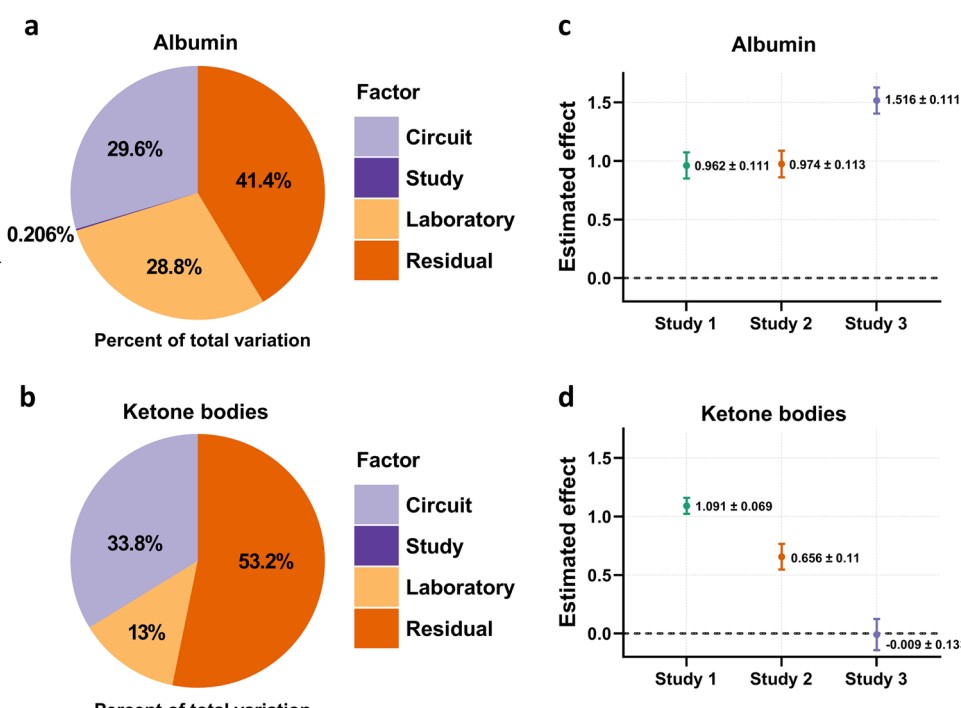

Chip2 platform[40] to address possible variations caused by control units, operators, and incubators.

We evaluated intra-study and inter-study reproducibility with a statistical methodology recently proposed by Schurdak et al.[41]. These analyses confirmed that the on-chip readouts have overall acceptable or excellent reproducibility within and across studies. Notably, the key readout of the model, the GTT with glucose and insulin measurements, demonstrated excellent intra-study reproducibility, ensuring high reliability in the model's primary context of use. The inter-study reproducibility of ketone-body measurement was poor in the healthy condition. Similar to the findings reported by Schurdak et al.[41], the poor reproducibility status was the result of one outlying study, potentially due to differences in cell source (different HepaRG batches) or bioanalysis methods (both fresh and frozen samples used). Furthermore, we demonstrated method's effectiveness in three pancreatic islet donors, enabling the investigation of varying susceptibilities for beta-cell damage and adaptation, which is challenging to achieve in animal models due to their monogenetic background. To accommodate studies on inter-donor variability also for the liver compartment, we are currently developing a pancreas-liver MPS with primary human hepatocytes.

Our current MPS focuses on liver-pancreas axis—a portion of insulin resistance which is a complex multi-organ disease. Including other target tissues for insulin action, such as an adipose-tissue model, would be valuable. While the liver plays a central role in controlling the glucose metabolism, adipose tissue regulates glucose and lipid metabolism by releasing free fatty acids, adipokines, and proinflammatory cytokines[53]. However, translational in vitro models of adipose tissue are not trivial to establish[54], especially because the adipose tissue is highly heterogeneous[55]. Recently, Slaughter et al. successfully coupled liver and adipose tissue models on chip with functional adipokine signaling for 14 days[56] and Tanataweethum et al. demonstrated the first insulin resistant liver-adipose (white and brown) MPS[57]. Expanding to a pancreas-liver-adipose MPS would broadly reflect insulin resistance pathophysiology and enable investigations of emerging therapies targeting adipose tissue[58]. Additionally, incorporating immune cells[59] and using hiPSC-derived organ models could further increase physiological relevance, reflecting the highly heterogenous disease progression and patient-specific pathophysiology. Indeed, MPS with patient-derived

cells hold a great potential for personalized medicine, including disease modeling of rare genetic diseases and selecting personalized drug treatments[60]. The pancreas-liver MPS, composed of human cells, holds even promise for studying new drug modalities such as oligonucleotide therapeutics. Generally, oligonucleotide therapeutics are specific for human gene sequence with limited homology to non-clinical species. Transgenic mice models or parallel development of species-specific oligonucleotide therapeutics are used to overcome the challenge with non-human pre-clinical models[9]. This MPS model could be used for studying efficacy, safety, targeting efficiency, and off-target effects of oligonucleotide therapeutics, similar to the recent use of a kidney MPS platform for assessing antisense oligonucleotide safety profiles[50,51].

In conclusion, the described pancreas-liver MPS reflects glucocorticoid-induced diabetes with impaired glucose regulation and islet dysfunction when treated with hyperglycemia and high HCT. Conversely, the model effectively maintains glucose homeostasis under physiological culture conditions with normoglycemia and low HCT. We have also shown that a partnership of MPS and in silico computing is useful for studies on multisystem diseases with complex organ-to-organ communication loops. Through testing in two laboratories, we have confirmed good intra-study and inter-study reproducibility, affirming the potential of the model to advance drug discovery and development. The human-cell-based pancreas-liver MPS serves as a preclinical platform for investigating disease mechanisms, identifying targets, and evaluating candidate drugs.

## Materials and methods
### Liver spheroid formation
All cell cultures were maintained at 37 °C and 5% $CO_2$ and conducted according to good cell culture practice[61]. We used terminally differentiated human HepaRG cells as a hepatocyte model as their gene expression profiles, regulatory pathways, functional glucose machinery and lipid metabolism are similar to that in primary human hepatocytes[62–64]. Furthermore, a functional insulin responsiveness was described for HepaRG cells[63] which is further improved in a three-dimensional spheroid culture[12]. Before liver spheroid formation, differentiated HepaRG hepatocyte-like cells (HPR116080, Biopredic, Lots HPR116NS080003, HPR116239-TA08 and HPR116222-TA08 or NSHPRG, Lonza, Lot HNS1014) were pre-cultured

**Table 1 | Intra-study reproducibility of the pancreas-liver MPS**

| On-chip readout | Study | # of circuits | # of time points | Max CV | ICC | Reproducibility status |
|---|---|---|---|---|---|---|
| **Healthy condition** | | | | | | |
| Albumin | 1 | 10 | 7 | 25.2 | 0.805 | Excellent (ICC) |
| | 2 | 8 | 7 | 17.1 | 0.743 | Acceptable (ICC) |
| | 3 | 8 | 7 | 26.3 | 0.349 | Acceptable (ICC) |
| Ketone bodies | 1 | 8 | 2 | 7.6 | 0.422 | Acceptable (ICC) |
| | 2 | 8 | 2 | 13.2 | 0.464 | Acceptable (ICC) |
| | 3 | 8 | 2 | 15.4 | 0.355 | Acceptable (ICC) |
| IL-1R2 | 2 | 4 | 3 | 46.3 | 0.969 | Excellent (ICC) |
| | 3 | 4 | 3 | 49.9 | 0.665 | Acceptable (ICC) |
| GTT Insulin | 1 | 4 | 4 | 9.4 | 0.908 | Excellent (ICC) |
| | 2 | 4 | 4 | 2.9 | 0.960 | Excellent (ICC, CV) |
| GTT Glucose | 1 | 4 | 4 | 20.0 | 0.953 | Excellent (ICC) |
| | 2 | 4 | 4 | 61.6 | 0.927 | Excellent (ICC) |
| **Diseased condition** | | | | | | |
| Albumin | 1 | 6 | 7 | 36.5 | 0.581 | Acceptable (ICC) |
| | 2 | 4 | 7 | 37.1 | 0.479 | Acceptable (ICC) |
| | 3 | 4 | 7 | 20.7 | 0.890 | Excellent (ICC) |
| Ketone bodies | 1 | 6 | 2 | 16.3 | 0.843 | Excellent (ICC) |
| | 2 | 4 | 2 | 38.2 | 0.459 | Acceptable (ICC) |
| | 3 | 4 | 2 | 38.0 | 0.294 | Acceptable (ICC) |
| IL-1R2 | 2 | 4 | 3 | 14.5 | 0.939 | Excellent (ICC) |
| | 3 | 4 | 3 | 7.3 | 0.977 | Excellent (ICC) |
| GTT Insulin | 1 | 4 | 4 | 3.5 | 0.724 | Excellent (CV) |
| | 2 | 4 | 4 | 5.1 | 0.882 | Excellent (ICC) |
| GTT Glucose | 1 | 4 | 4 | 50.6 | 0.905 | Excellent (ICC) |
| | 2 | 4 | 4 | 48.1 | 0.946 | Excellent (ICC) |

Statistical assessment of the reproducibility is based on intra-class correlation coefficient (ICC) and the maximum coefficient of variation (CV). Reproducibility is classified as "Excellent" if Max CV ≤ 5% or ICC ≥ 0.8; "Acceptable" if 5% < Max CV < 15% or 0.2 ≤ ICC < 0.8; or "Poor" if ICC < 0.2. Studies 1 and 2 were performed at TissUse and Study 3 at AstraZeneca.

for 4 days as previously described with a modification to medium composition[12]. Glucose and insulin concentration of the pre-culture medium were adjusted to physiological levels resulting in the following composition: Williams' medium E (P04-29050S4, PAN-Biotech, w/o glucose, w/o L-glutamine, w/o phenol red) supplemented with 10% fetal bovine serum (FBS; 35-079-CV, Corning or 10270-106, Gibco), 5.5 mM glucose (25-037-CIR, Corning or 072397, Fresenius Kabi), 1 nM insulin (P07-4300, PAN-Biotech or 12585-014, Gibco), 2 mM GlutaMax (35050-061, Gibco), 50 µM hydrocortisone hemisuccinate (H4881, VWR or H2270, Sigma-Aldrich), 50 µg/ml gentamycin sulfate (30-005-CR, Corning or 15710-049,

**Table 2 | Inter-study reproducibility of the pancreas-liver MPS**

| On-chip readout | Studies | # of circuits | # of time points | Max CV | ICC | Reproducibility status |
|---|---|---|---|---|---|---|
| **Healthy condition** | | | | | | |
| Albumin | 1,2,3 | 26 | 7 | 34.6 | 0.277 | Acceptable (ICC) |
| Ketone bodies | 1,2,3 | 24 | 2 | 32.1 | 0 | Poor (CV) |
| IL-1R2 | 2,3 | 8 | 3 | 54.5 | 0.739 | Acceptable (ICC) |
| GTT Insulin | 1,2 | 8 | 4 | 92.8 | 0.499 | Acceptable (ICC) |
| GTT Glucose | 1,2 | 8 | 4 | 7.1 | 0.861 | Excellent (ICC) |
| **Diseased condition** | | | | | | |
| Albumin | 1,2,3 | 14 | 7 | 40 | 0.553 | Acceptable (ICC) |
| Ketone bodies | 1,2,3 | 14 | 2 | 30 | 0.566 | Acceptable (ICC) |
| IL-1R2 | 2,3 | 8 | 3 | 24 | 0.924 | Excellent (ICC) |
| GTT Insulin | 1,2 | 8 | 4 | 89.6 | 0.384 | Acceptable (ICC) |
| GTT Glucose | 1,2 | 8 | 4 | 4.8 | 0.753 | Excellent (CV) |

Statistical assessment of the reproducibility is based on the intra-class correlation coefficient (ICC) and the maximum coefficient of variation (CV). Reproducibility is classified as "Excellent" if Max CV ≤ 5% or ICC ≥ 0.8; "Acceptable" if 5% < Max CV < 15% or 0.2 ≤ ICC < 0.8; or "Poor" if ICC < 0.2. Studies 1 and 2 were performed at TissUse and Study 3 at AstraZeneca.

Gibco) and 0.25 µg/ml amphotericin B (30-003-CF, Corning). Differentiated HepaRG cells were thawed and seeded confluently in pre-culture medium containing 0.5% DMSO. One day after thawing, medium was exchanged with pre-culture medium containing 2% DMSO. The cells were maintained in this medium for three days until spheroid formation.

Primary human hepatic stellate cells (HHSteC, S00354, BioIVT, Lot PFP) were expanded in Stellate Cell Medium (5301, ScienCell) supplemented with Stellate Cell Growth Supplement, 2% FBS and 1% penicillin/streptomycin, and cryopreserved in FBS with 10% DMSO (23500.297, VWR). The HHSteCs (P3-4) were thawed at least two days before spheroid formation and pre-cultured in stellate cell medium until spheroid formation.

Liver spheroids were formed for 3 days in 384-well spheroid microplates (3830, Corning) with 24,000 differentiated HepaRG hepatocytes and 1000 HHSteCs per spheroid as described previously[12]. Briefly, differentiated HepaRG cells and HHSteCs were collected from their culture vessel using 0.25%/2.21 mM or 0.05%/0.53 mM trypsin/EDTA respectively. Trypsin was neutralized with pre-culture medium (without DMSO) and the cells were pelleted by centrifugation (HepaRG: $300 \times g$ for 5 min, HHSteC: $80 \times g$ for 5 min). Pellets were resuspended in pre-culture medium and cell concentrations were determined. Cell suspensions were combined and the volume was adjusted with pre-culture medium (without DMSO) in order to obtain a HepaRG concentration of 480,000 cells/ml and a HHSteC concentration of 20,000 cells/ml. Fifty µl of this HepaRG/HSteC cell suspension was loaded into each well of the 384-well spheroid microplate. The plate was centrifuged for 2 min at $200 \times g$ and incubated at 37 °C and 5% CO$_2$ for 3 days. Once compact spheroids had formed, 40 spheroids were collected into a 24-well ultra-low attachment plate (3473, Corning) for each co-culture replicate, and incubated in 1 ml pre-culture medium (without DMSO) overnight on a 3D rotator (PS-M3D; Grant-bio) before transfer to the islet-liver co-culture.

**Pre-culture of pancreatic islets**
We used commercially available human pancreatic islet microtissues (MT-04-002-0, InSphero) as a pancreatic islet model. The microtissues are manufactured from a dissociated human pancreatic islet suspension and have a defined cell number. After arrival, the pancreatic islet microtissues (here called islets) were maintained for 5 days in Akura™ 96 Spheroid Microplate (CS-09-004-01, InSphero) according to the manufacturer's instructions. Medium was exchanged every 2–3 days with 70 µl of Human Islet Maintenance Medium (CS-07-005-02; InSphero). The used pancreatic islet donors are listed in Supplementary Table 2.

**Pancreas-liver MPS**
Experimental design of the work is summarized in Supplementary Table 1. We performed co-cultures with islet and liver spheroids on a commercially available multi-organ-chip HUMIMIC Chip2 (TissUse) platform (Fig. 1d). This MPS has two culture compartments for the integration of spatially separated organ models. The culture compartments are interconnected by a microfluidic channel which enables re-circulation of the common co-culture medium. An on-chip micropump drives a pulsatile flow supporting long-term perfusion and communication between the organ models. Design and fabrication of the Chip2 were described previously[65,66].

Three days before insertion of the organ models, the chips were prepared for cultivation by replacing the storage buffer with 300 µl co-culture medium in each culture compartment (total volume per circulation was 605 µl). The chips were connected via air tubes to the control unit (HUMIMIC Starter) operating the on-chip micropump. The control unit was set to 0.45 Hz, 500 mbar pressure and –500 mbar vacuum resulting in an average volumetric flow rate of 4.94 µl/min between the culture compartments.

On the day of organ model transfer, the liver spheroids were washed twice with phosphate-buffered saline (PBS) to remove insulin from pre-culture medium. Subsequently, the liver spheroids were equilibrated for at least 2 h in an insulin-free co-culture medium composed of Williams' medium E (w/o glucose, w/o L-glutamine, w/o phenol red), 10% FBS, 2 mM GlutaMax, 50 µg/ml gentamycin sulfate, and 0.25 µg/ml amphotericin B. Glucose concentration was 5.5 mM in the normoglycemic condition and 11 mM in the hyperglycemic condition, and hydrocortisone concentration was either 10 nM or 50 µM (indicated in each study and condition). The

https://doi.org/10.1038/s42003-024-06514-w **Article**

islets were similarly equilibrated in the co-culture medium for at least 2 h. After equilibration, 40 liver spheroids and 10 pancreatic islets were transferred to their respective culture compartment with 300 μl of fresh co-culture medium. Liver spheroids were collected using a wide-bore filter tip (T-205-WB-C-R-S, Corning), and carefully transferred into the liver compartment. In parallel, 10 islets were collected into a 1.5 ml microtube, pelleted by a brief centrifugation (1 min, 200 × g) and transferred to the islet compartment. Alternatively, the islets were collected using an electronic single-channel pipette (Xplorer plus, Eppendorf) and directly transferred into the chips. The chips were reconnected to the control units and incubated at 37 °C and 5% CO₂. The co-culture medium in both culture compartments was exchanged completely after 24 h (adaptation time to the dynamic culture) and subsequently every 48 h for a total co-culture duration of 15–16 days. In studies 2 and 3, medium was supplemented with 10 μM EdU for proliferation analysis for the last 5 days of culture. In studies 1, 2, and 3, some islets were statically cultured in Akura™ 96 Spheroid Microplate and 70 μl medium was exchanged on day 1, 3, 5, 7, 9, 11 and 13 to mimic the co-cultures medium exchange regime.

### Hydrocortisone dose-response

Islets were cultured in Akura™ 96 Spheroid Microplate in 70 μl of normoglycemic co-culture medium containing 0, 0.05 nM, 0.5 nM, 5 nM, 50 nM, 500 nM, 5 μM and 50 μM HCT. Medium with HCT was renewed on day 1, 3, 5, 7, 9, 11 and 13 to mimic the co-cultures medium exchange regime. After 15 days of culture, the islets were analyzed for their glucose-stimulated insulin secretion as described below.

### IL-1R2 treatment

Islets were cultured in Akura™ 96 Spheroid Microplate in 70 μl of Human Islet Maintenance Medium and they were treated with 0.3 ng/ml or 30 ng/ml of human recombinant IL-1R2 protein (10111-H08H, Sino Biological) for 16 days. Medium with IL-1R2 was renewed three times a week. For the last five days of culture, medium was also supplemented with 10 μM EdU for proliferation analysis. Islets cultured in insulin-free chip co-culture medium with 10 nM hydrocortisone and 11 mM glucose (Low HCT-HG condition) served as a positive control since this medium was shown to significantly induce islet proliferation (Supplementary Fig. 7). After finishing the culture, the islets were analyzed for their glucose-stimulated insulin secretion and proliferation using EdU incorporation assay as described below.

### Glucose tolerance test

GTT assays were performed as described previously[12] at different timepoints during the co-culture. Briefly, the co-culture medium was exchanged in both culture compartments with a co-culture medium containing 11 mM glucose (−300 μl, +315 μl). Fifteen μl of supernatant samples were collected at 0, 8, 24, and 48 h to monitor glucose and insulin concentrations. To obtain sufficient sample volumes for the analysis, samples from the liver and islet compartments were pooled. For optimal sample recovery, samples were stored in 96-well PCR plates (30133358, Eppendorf) and sealed using aluminum foil to minimize evaporation during storage. Samples were stored at −80 °C until glucose and insulin measurements (see "Analysis of soluble markers").

### Glucose-stimulated insulin secretion

To assess functionality of the islets after the co-culture, islets from the chips were extracted, transferred into Akura™ 96 Spheroid Microplate and a GSIS assay was performed on individual islets. The islets were first washed twice with 70 μl of Krebs-Ringer solution containing 2.8 mM glucose (low-glucose solution), followed by equilibration in 70 μl of low-glucose solution for 1–2 h. Next, the islets were washed twice with 70 μl of low-glucose solution and incubated for 2 h in 50 μl of low-glucose solution to measure basal insulin secretion. Following this, the islets were washed once with 70 μl of Krebs-Ringer solution containing 16.8 mM glucose (high-glucose solution) and subsequently incubated in 50 μl of high-glucose solution for 2 h to measure the glucose-stimulated insulin

secretion. Basal and glucose-stimulated samples were collected after incubations and stored at −80 °C until insulin measurement.

### Analysis of soluble markers

Cell culture supernatant collected during medium exchanges were analyzed for albumin (10242, Diagnostic Systems) and ketone bodies (3-beta-hydroxybutyrate, Autokit 3-HB, Fujifilm Wako) on an Indiko Plus chemical analyzer (Thermo Fisher Scientific) according to the manufacturer's instructions. IL-1R2 concentrations were determined in culture supernatants by an ELISA assay (EHIL1R2, Thermo Scientific) according to the manufacturer's instructions. Samples taken during the GTT were analyzed for glucose (1070-500, Stanbio Laboratory) and insulin (10-1113-10, Mercodia) according to the manufacturer's instructions.

### Computational models

#### Hypothesis testing using computational modeling.

We used mathematical modeling as a tool to test mechanistic hypotheses on experimental data. A mechanistic hypothesis corresponds to a formulation of causal mechanisms key to produce the observed behavior in the data. Hypothesis testing via mathematical modeling is an iterative approach (Fig. 2b).

In the first step, the existing hypotheses are translated into a set of mathematical equations (i.e., corresponding mathematical models). We considered two hypotheses for the observed glucose and insulin responses in the pancreas-liver MPS: H1, "Insulin resistance is caused by hyperglycemia alone", and H2, "Insulin resistance is caused by a combination of hyperglycemia and an additional diabetogenic factor".

The second step involves the acquisition of experimental data and fitting the mathematical models to these data by optimization of the model parameters. The hypotheses are initially evaluated based on the outcome of this optimization. If the mathematical model cannot provide an acceptable agreement with the data, according to statistical analyses, then the corresponding hypothesis is rejected and must be revised. On the other hand, if the model can provide an acceptable agreement with data, the corresponding hypothesis is not rejected. The non-rejected models can then be used to generate uniquely identified predictions with uncertainty[67], that allow for designing new experiments that could distinguish between the remaining hypotheses. The experiments are performed, and the predictions are compared against the new experimental data. If the model predictions agree with the experimental data, the corresponding hypothesis is accepted. On the contrary, if the predictions do not agree with the experimental data, the model is rejected and a new iteration in the hypothesis testing cycle is performed. Several iterations can be performed until a final model has been found. In the following, we describe the mathematical model with its equations, and the hypothesis testing procedure in detail.

#### A computational model for glucose metabolism in the pancreas-liver MPS.

We used our previously developed computational model[14] as a basis to implement the hypotheses studied in this paper. The model describes glucose metabolism in the pancreas-liver MPS (Supplementary Fig. 2b). More specifically, it describes crucial biological processes underlying glucose regulation on a short-term basis (meal response), as well as long-term changes in physiological variables related to impaired glucose homeostasis, such as insulin resistance and beta-cell adaptation. This model was constructed based on experimental data from seven independent studies corresponding to seven different islet donors[14].

The computational model is formulated using ordinary differential equations (ODEs), which have the following general structure:

$$\frac{d}{dt}\vec{x}(t) = f\left(\vec{x}(t), \vec{p}, \vec{u}(t)\right)$$
$$\vec{x}(0) = \vec{x}_0$$
$$\vec{y}(t) = g\left(\vec{x}(t), \vec{p}, \vec{u}(t)\right)$$

where $\vec{x}(t)$ is the state vector describing the dynamics of concentrations or amounts and $\vec{p}$ are the parameters, which here correspond to kinetic rate constants. $\vec{u}(t)$ is a vector containing the external inputs. $\vec{x}(0)$ contains the initial conditions, i.e., the values of the states at $t = 0$. The vector $\vec{y}(t)$ represents the simulated model outputs, which correspond to the measured experimental signals. Finally, $f$ and $g$ are non-linear smooth functions that describe a set of mechanistic assumptions.

**Derivation of the computational model.** The computational model is based on the interplay between two components corresponding to different time scales: fast (hours) and slow (weeks). The fast model describes glucose and insulin dynamics between medium exchanges, which take place every 48 h. The slow model describes the dynamics of long-term variables representing disease progression, such as the development of insulin resistance in the liver spheroids and beta-cell adaptation in the islets. The interplay between these two models allows short-term variables to impact long-term disease progression (e.g., impact of daily glucose levels on insulin resistance and beta-cell volume) and vice versa. The model includes two compartments, each of them representing a specific culture compartment in the MPS (liver or pancreas) comprising a corresponding organ model and co-culture medium. The compartments are connected in a closed loop, with circulating medium determined by a flow rate parameter. The model equations are described in detail previously[14] and summarized below.

Glucose content in the co-culture medium within the liver compartment varies with glucose dosing to the system, endogenous glucose production and glucose uptake by the liver spheroids, as well as glucose inflow from and outflow to the pancreas compartment:

$$\frac{dNG_{m,\text{liver}}(t)}{dt} = G_d(t) + V_{\text{HepaRG,spheroids}} \cdot \text{EGP}(t)$$
$$- V_{\text{HepaRG,spheroids}} \left( E_{G0} + S_I(t) \cdot \frac{NI_{m,\text{liver}}(t)}{V_{m,\text{liver}}} \right) \frac{NG_{m,\text{liver}}(t)}{V_{m,\text{liver}}}$$
$$+ Q \cdot \frac{NG_{m,\text{pancreas}}(t)}{V_{m,\text{pancreas}}} - Q \cdot \frac{NG_{m,\text{liver}}(t)}{V_{m,\text{liver}}} \left( \frac{\text{mmol}}{\text{h}} \right)$$

where $NG_{m,\text{liver}}(t)$ and $NG_{m,\text{pancreas}}(t)$ are the number of glucose molecules (mmol) in the culture medium corresponding to the liver and pancreas compartments, respectively, and $NI_{m,\text{liver}}(t))$ is the number of insulin molecules in the co-culture medium within the liver compartment (mIU). The glucose input rate $G_d(t)$ (mmol/h) defines glucose variations due to media exchanges, and $\text{EGP}(t)$ describes endogenous glucose production in the liver spheroids (mmol/L/h). $\text{EGP}(t)$ was set to zero based on the observed decline in glucose levels below normoglycemia (5.5 mM) in our system. Glucose uptake by the liver spheroids accounts for both insulin-independent uptake, determined insulin-independent glucose disposal rate $E_{G0}$ (1/h), and an insulin-dependent uptake regulated by the insulin sensitivity of the liver spheroids $S_I(t)$ (L/mIU/h). The parameters describing the flow rate between culture compartments ($Q$ (L/h)), the total volume of HepaRG cells in the liver spheroids ($V_{\text{HepaRG,spheroids}}$ (L)) and the volume of co-culture medium in the liver and pancreas compartments ($V_{m,\text{liver}}$ and $V_{m,\text{pancreas}}$ (L), respectively) account for the operating conditions in the MPS.

In the computational model, insulin sensitivity of the liver spheroids $S_I(t)$ decreases progressively from its initial value at the beginning of the co-culture $S_{I0}$ (L/mIU/h), as the liver spheroids are exposed to hyperglycemic concentrations (i.e., above normoglycemia) over time:

$$S_I(t) = S_{I0} \cdot \left( 1 - \frac{I_{\max,Si} \cdot G_{\text{int}}(t)}{EC50_{Si} + G_{\text{int}}(t)} \right) (\text{L/mIU/h}).$$

This decrease is determined by the maximal fractional reduction $I_{\max,Si}$, and with half of the maximal fractional reduction occurring at $EC50_{Si}$ (mmol*h/L). The hyperglycemic periods are quantified

by the integral of excess glucose $G_{\text{int}}(t)$, which represents the difference between the glucose concentration in the liver compartment $NG_{m,\text{liver}}(t)/V_{m,\text{liver}}$ (mmol/L) and a normoglycemic glucose concentration $G_{\text{normo}}$ (5.5 mmol/L):

$$\frac{dG_{\text{int}}(t)}{dt} = \begin{cases} \frac{NG_{m,\text{liver}}(t)}{V_{m,\text{liver}}} - G_{\text{normo}} & \frac{NG_{m,\text{liver}}(t)}{V_{m,\text{liver}}} - G_{\text{normo}} \geq 0 \\ 0 & \frac{NG_{m,\text{liver}}(t)}{V_{m,\text{liver}}} - G_{\text{normo}} < 0 \end{cases} (\text{mmol/L})$$

Glucose content in the pancreas compartment is described as:

$$\frac{dNG_{m,\text{pancreas}}(t)}{dt} = G_d(t) + Q \cdot \frac{NG_{m,\text{liver}}(t)}{V_{m,\text{liver}}} - Q \cdot \frac{NG_{m,\text{pancreas}}(t)}{V_{m,\text{pancreas}}} (\text{mmol/h})$$

Insulin content in the pancreas compartment depends on the release of insulin from beta cells in the islets, and insulin inflow from and outflow to the liver compartment. Insulin release from the beta cells was modeled as a combination of the volume of beta cells in the islets ($V_{\beta,\text{islets}}(t)$ (L)), the insulin secretion capacity per unit volume of beta cells (denoted $\sigma(t)$ (mIU/L/h)), and the glucose concentration resulting in half-of-maximum response to insulin (denoted $EC50_I$ (mmol/L)). The full equation describing insulin content in the pancreas compartment then becomes:

$$\frac{dNI_{m,\text{pancreas}}(t)}{dt} = V_{\beta,\text{islets}}(t) \cdot \sigma(t) \cdot \frac{\left( \frac{NG_{m,\text{pancreas}}(t)}{V_{m,\text{pancreas}}} \right)^2}{EC50_I^2 + \left( \frac{NG_{m,\text{pancreas}}(t)}{V_{m,\text{pancreas}}} \right)^2}$$
$$+ Q \cdot \frac{NI_{m,\text{liver}}(t)}{V_{m,\text{liver}}} - Q \cdot \frac{NI_{m,\text{pancreas}}(t)}{V_{m,\text{pancreas}}} (\text{mIU/h})$$

where $NI_{m,\text{pancreas}}(t)$ and $NI_{m,\text{liver}}(t)$ are the number of insulin molecules (mIU) in the pancreas and the liver compartment, respectively.

Furthermore, the insulin secretion capacity of the beta cells was modeled as a decreasing function of time, determined by the parameter $\alpha$ (h²):

$$\sigma(t) = \sigma_{\max} \cdot \left( 1 - \frac{t^2}{\alpha + t^2} \right) (\text{mIU/L/h})$$

where $\sigma_{\max}$ (mIU/L/h) represents the maximal insulin secretion rate of the beta cells (i.e., at the beginning of the co-culture).

The variable $V_{\beta,\text{islets}}(t)$ (L) describes the changes in volume of beta cells in the pancreatic islets over the co-culture time, according to the following equation:

$$\frac{dV_{\beta,\text{islets}}(t)}{dt} = k_v(-d_0 + r_1 G_{\text{slow,pancreas}}(t) - r_2 G_{\text{slow,pancreas}}(t)^2) \cdot V_{\beta,\text{islets}}(t) (\text{L/h})$$

where $d_0$ is the death rate at zero glucose (h⁻¹) and $r_1 = r_{1,r} + r_{1,a}$ (L/mmol/h) and $r_2 = r_{2,r} + r_{2,a}$ (L²/mmol²/h), where $r_{1,r}, r_{1,a}$ (L/mmol/h), $r_{2,r}, r_{2,a}$ (L²/mmol²/h) are parameters that determine the dependence on glucose of the replication and apoptosis rates. The dimensionless parameter $k_v$ was introduced to account for potential differences in behavior between islets in our in vitro system and rodent islets in the model of Topp et al.

The variable $G_{\text{slow,pancreas}}(t)$ (mmol/L) represents the long-term average (i.e., daily) glucose concentration in the co-culture medium as given by:

$$\frac{dG_{\text{slow,pancreas}}(t)}{dt} = \frac{G_{\text{pancreas}}(t) - G_{\text{slow,pancreas}}(t)}{\tau_{\text{slow}}} (\text{mmol/L/h})$$

where the parameter $\tau_{\text{slow}}$ (h) represents the time scale of the averaging process.

Insulin content in the liver compartment decreases over time according to the hepatic insulin elimination rate constant $k_{\text{elimination }I,\text{spheroids}}$ (1/h):

$$\frac{dNI_{m,\text{liver}}(t)}{dt} = Q \cdot \frac{NI_{m,\text{pancreas}}(t)}{V_{m,\text{pancreas}}} - Q \cdot \frac{NI_{m,\text{liver}}(t)}{V_{m,\text{liver}}}$$
$$- V_{\text{HepaRG,spheroids}} \cdot k_{\text{elimination }I,\text{spheroids}} \cdot \frac{NI_{m,\text{liver}}(t)}{V_{m,\text{liver}}} (\text{mIU/h})$$

The concentrations of glucose and insulin in each compartment were calculated by dividing the insulin and glucose content, respectively, by the volume of co-culture medium in the compartment:

$$G_{\text{liver}}(t) = \frac{NG_{m,\text{liver}}(t)}{V_{m,\text{liver}}} (\text{mmol/L})$$

$$G_{\text{pancreas}}(t) = \frac{NG_{m,\text{pancreas}}(t)}{V_{m,\text{pancreas}}} (\text{mmol/L})$$

$$I_{\text{liver}}(t) = \frac{NI_{m,\text{liver}}(t)}{V_{m,\text{liver}}} (\text{mIU/L})$$

$$I_{\text{pancreas}}(t) = \frac{NI_{m,\text{pancreas}}(t)}{V_{m,\text{pancreas}}} (\text{mIU/L})$$

Glucose and insulin samples in the co-culture studies were obtained by pooling samples from both the liver and the pancreas compartment. Therefore, the resulting glucose and insulin measurements ($G(t)$ and $I(t)$, respectively), were computed as:

$$G(t) = G_{\text{liver}}(t) \cdot V_{\text{sample,liver}} + G_{\text{pancreas}}(t) \cdot \frac{V_{\text{sample,pancreas}}}{(V_{\text{sample,liver}} + V_{\text{sample,pancreas}})} (\text{mmol/L})$$

$$I(t) = I_{\text{liver}}(t) \cdot V_{\text{sample,liver}} + I_{\text{pancreas}}(t) \cdot \frac{V_{\text{sample,pancreas}}}{(V_{\text{sample,liver}} + V_{\text{sample,pancreas}})} (\text{mIU/L})$$

where $V_{\text{sample,liver}}$ and $V_{\text{sample,pancreas}}$ are the volumes of co-culture media collected from the liver and pancreas compartment, respectively, in each sample (15 µl).

The initial conditions for the model states are listed below:

$$NG_{m,\text{liver}}(0) = (G_{\text{dose}} + \Delta G_{d1}) \cdot V_{m,\text{liver}} (\text{mmol})$$
$$NG_{m,\text{pancreas}}(0) = (G_{\text{dose}} + \Delta G_{d1}) \cdot V_{m,\text{islets}} (\text{mmol})$$
$$NI_{m,\text{liver}}(0) = \Delta I_{d1} \cdot V_{m,\text{liver}} (\text{mIU})$$
$$NI_{m,\text{pancreas}}(0) = \Delta I_{d1} \cdot V_{m,\text{pancreas}} (\text{mIU})$$
$$t(0) = 0 \,(\text{h})$$
$$G_{\text{int}}(0) = 0 \,(\text{mmol} \cdot \text{h/L})$$
$$G_{\text{slow,pancreas}}(0) = 5.5 \,(\text{mmol/L})$$
$$V_{\beta,\text{islets}}(0) = 8.8 \cdot 10^{-9} (\text{L})$$

where $\Delta G_{d1}$ (mmol/L), $\Delta I_{d1}$ (mIU/L) are offset parameters that account for experimental errors related to the medium exchange performed on day 1. The experimental errors in the glucose concentration at 0 h can be due to varying co-culture medium volumes in the culture compartments, varying glucose concentration in the co-culture medium, or glucose assay-dependent variations. Values of insulin concentration different from zero at $t = 0$ h could be due to co-culture medium remaining in the chip (both in the culture compartments and the microfluidic channel) during the medium exchange corresponding to the first GTT. Similarly, the model parameters $(\Delta G_{d13}, \Delta I_{d13})$ account for errors in glucose and insulin concentrations, respectively, during the medium exchanges performed on day 13.

**Hypothesis testing to unravel the origin of insulin resistance in the pancreas-liver MPS.** We tested two hypotheses that could explain the glucose and insulin responses observed in the MPS (Fig. 1e). The first hypothesis (H1) assumes that insulin resistance is caused by hyperglycemia alone, while the second hypothesis (H2) assumes that insulin resistance is caused by hyperglycemia and an additional diabetogenic factor. The model described in Casas et al.[14] implements hypothesis H1. Therefore, we created a second computational model implementing

hypothesis H2, by including an equation to model the effect of an additional diabetogenic factor on insulin sensitivity. This effect was modeled as a sigmoidal function of time, with maximal fractional reduction $I_{\text{max,additional}}$, and with half of the maximal fractional reduction occurring at $EC50_{\text{additional}}$ (mmol h/L):

$$S_I(t) = S_{I0} \cdot \left(1 - \frac{I_{\text{max},Si} \cdot G_{\text{int}}(t)}{EC50_{Si} + G_{\text{int}}(t)}\right) \cdot \left(1 - \frac{I_{\text{max,additional}} \cdot t^2}{EC50_{\text{additional}}^2 + t^2}\right)$$

Each computational model was calibrated against the experimental data of glucose and insulin from the pancreas-liver MPS. To perform this calibration, the model parameters were estimated using nonlinear optimization, by finding parameter values that provided an acceptable agreement with the experimental data according to the following cost function:

$$V(p) = \sum_i \sum_t \frac{(y_i(t) - \hat{y}_i(t, p))^2}{\text{SEM}_i(t)^2}$$

where $i$ is summed over the number of experimental time-series for the given experiment $y_i(t)$ and $\hat{y}_i(t, p)$ represents the model simulations and $p$ the model parameters. SEM denotes the standard error of the mean and $t$ the measured time points in each time-series. To handle uncertainty in the estimation, we used a simulated annealing approach[68] to find the set of acceptable parameters that provided an acceptable agreement with the experimental data according to a statistical $\chi^2$ test[67,69] with a significance level of 0.05.

We found a good visual agreement with the experimental data for both models corresponding to H1 and H2 (Fig. 2d, e). This visual agreement was statistically supported by the fact that both models passed a $\chi^2$ test at a significance level $\alpha = 0.05$, with a value of the cost for the optimal parameter set $p_{\text{opt}}$ lower than the $\chi^2$-threshold ($V(p_{\text{opt},H1}) = 21.62 < 37.65$, $V(p_{\text{opt},H2}) = 28.32 < 37.65$).

To be able to discriminate between H1 and H2, we performed predictions of glucose and insulin responses for different doses of added insulin to the co-culture medium, and selected an insulin dose that would provide detectable differences between the glucose responses for these hypotheses (i.e., differences larger than the average SEM across samples in the experimental data). The model predictions were made for the entire set of acceptable parameters. To visualize these predictions, we simulated model responses for the maximal and minimal values of each parameter within the set of acceptable parameters. We then calculated the boundaries of the prediction by computing the maximal and minimal value of the prediction for each time point and visualized the area between these boundaries (Fig. 2d). We performed the corresponding experiments for the calculated insulin dose (23 nM) and computed the model prediction. No acceptable agreement with the experimental data was found for H1, and this hypothesis was therefore rejected (Fig. 2d). H2, on the other hand, showed good visual agreement with the experimental data (Fig. 2e), which was also confirmed with a $\chi^2$ test at significance level $\alpha = 0.05$ ($V(p_{\text{opt},H2}) = 3.45 < 12.59$).

**Simulating the effect HCT concentration in the pancreas-liver MPS.** In the computational model, the effect of high HCT on the pancreas-liver MPS was modeled as a decrease in both the insulin sensitivity of the liver spheroids $S_I(t)$ and the insulin secretion capacity of the $\beta$ cells $\sigma(t)$ over time, as follows:

$$S_I(t) = S_{I0} \cdot \left(1 - \frac{I_{\text{max},Si} \cdot G_{\text{int}}(t)}{EC50_{Si} + G_{\text{int}}(t)}\right) \cdot \left(1 - \frac{I_{\text{max,additional}} \cdot t^2}{EC50_{\text{additional}}^2 + t^2}\right)$$
$$\sigma(t) = \sigma_{\text{max}} \cdot \left(1 - \frac{t^2}{\alpha + t^2}\right) (\text{mIU/L/h})$$

where the term in $S_I(t)$ with two additional parameters $I_{\text{max,additional}}$ and $EC50_{\text{additional}}$ represents the effect of high HCT $S_I(t)$.

To model the effect of physiological HCT levels on the pancreas-liver MPS, we omitted the terms corresponding to these decreases in $S_I(t)$ and

$\sigma(t)$, leading to the following equations:

$$S_I(t) = S_{I0} \cdot \left(1 - \frac{I_{\max,Si} \cdot G_{\mathrm{int}}(t)}{EC50_{Si} + G_{\mathrm{int}}(t)}\right)$$
$$\sigma(t) = \sigma_{\max}$$

To predict the glucose and insulin responses in the pancreas-liver MPS under physiological HCT concentrations, we first calibrated the computational model using experimental data under high HCT levels from two GTT experiments; one GTT starting at day 1 (GTT day 1–3) and one GTT starting at day 13 (GTT day 13–15) (Fig. 4a, high HCT). With the optimal parameter values obtained from this estimation as a start guess, we then optimized the parameters representing insulin sensitivity at the beginning of the co-culture $S_{I0}$ and the insulin secretion capacity of the beta cells $\sigma_{\max}$ using data under physiological HCT levels from another GTT starting at day 1 (Fig. 4b, low HCT). This optimization was done to establish a baseline for $S_{I0}$ and $\sigma_{\max}$ to predict the glucose and insulin responses. We then used this parameter set to predict the glucose and insulin responses under physiological HCT levels. In doing so, we omitted the decreases in $S_I(t)$ and $\sigma(t)$ over time, as previously described.

**Data pre-processing**. Given the small number of replicate platforms in the MPS studies (2–6), we assume that the SEM values measured experimentally are an underestimation of the true uncertainty in the data. We considered SEM values below 5% of the corresponding mean to be unrealistic and corrected for possible measurement errors by setting these SEM values to the largest measured SEM value across all data points in the experimental dataset. Furthermore, we accounted for experimental errors in glucose and insulin measurements due to media-exchanges by including measured offsets in concentrations at the beginning of GTTs ($t = 0$ within a given GTT) as an additional contribution to the total SEM for all data points corresponding to the given GTT. The resulting SEM values are given as error bars in all figures.

**Software**. Computations were carried out in MATLAB R2022b (The Mathworks Inc., Natick, Massachusetts, USA) using IQM tools (Inti-Quan GmbH, Basel, Switzerland) and the MATLAB Global Optimization toolbox, as well as in Python (v 3.9.13). Figures 1a–c, e, 2a–c and Supplementary Fig. 1 were created with BioRender.com.

### Gene expression analysis
**RNA isolation and quantitative real-time PCR**. After the co-culture, liver spheroids in the culture compartments were washed three times with PBS and the spheroids were removed using a sterile blunt end needle (9180117, B.Braun) for RNA isolation. Spheroids were transferred into PCR-clean 1.5 ml microtubes with 100 μl of lysis buffer (LB1 from Macherey-Nagel or 700 μl of Buffer RLT (79216, Qiagen). Lysates were snap-frozen and stored at −80 °C. RNA was isolated using the NucleoSpin® RNA Plus XS kit (740990.50, Macherey-Nagel) or RNeasy Mini Kit (74104, Qiagen). cDNA was synthesized using TaqMan® Reverse Transcription Kit (Thermo Fisher Scientific). Real-time PCR was performed using the SensiFAST SYBR Lo-ROX Kit (BIO-94020, Bioline) in a QuantStudio 5 Real-Time PCR System (Thermo Fisher Scientific). Primers are shown in Supplementary Table 3. Relative gene expression was determined using the comparative CT (ΔΔCt) method with TBP as endogenous control gene.

**RNA sequencing**. The quantity and quality of RNA samples was assessed using the standard sensitivity RNA fragment analysis kit on Fragment Analyzer (Agilent Technologies). All samples had an RNA integrity number >8 and were deemed of sufficient quantity and quality for RNA-seq analysis. Samples were diluted to a final quantity of 150 ng/sample of total RNA. The KAPA mRNA HyperPrep kit (Roche) was used for reverse transcription, generation of double stranded cDNA and subsequent library preparation and indexing to facilitate multiplexing

(Illumina TruSeq). All libraries were quantified with the Fragment Analyzer using the standard sensitivity NGS kit (Agilent Technologies), pooled in equimolar concentrations and quantified with a Qubit Fluorometer (Thermo Fisher Scientific) with the DNA HS kit (Thermo Fisher Scientific), the library pool was further diluted to 2.2 pM and sequenced at >20 M paired end reads/sample using the High Output regent kit to 150 cycles on an Illumina NextSeq500.

RNASeq data were analyzed using bcbio (version 1.1.0) and differential analysis was performed with DESeq2 (version 1.18.1).

### Proteomic analysis
**Sample preparation for proteomic analysis**. For proteomic analysis, the co-cultures were incubated in FBS-free co-culture medium for the last 4 days (d11–15). After finishing the culture, supernatants were collected from both pancreas and liver compartments and combined in a 1.5 ml microtube. Samples were first centrifuged at $300 \times g$ for 10 min at RT, to remove any remaining cells, and then supernatants were transferred into new tubes for centrifugation at $10,000 \times g$ for 10 min at 4 °C. The supernatants were stored at −80 °C until sample preparation for nano-scale liquid chromatographic tandem mass spectrometry (nLC-MS/MS) performed on two MPS media experiments and MS instruments, Q Exactive™ HF Orbitrap or Fusion™ Lumos™ Tribrid™ (Thermo Fisher Scientific).

**Sample preparation, peptide labeling and fractionation for Q Exactive™ HF analysis**. Equal volumes of the cell culture supernatants from each condition was concentrated on nanosep 10k omega filters (Pall Corporation, Port Washington, NY, USA) prerinsed with 50 mM triethylammonium bicarbonate (TEAB, Sigma-Aldrich) and was washed twice in the filter with 500 μL 50 mM TEAB, by spinning at $14,000 \times g$ for 20 min at 4 °C. Proteins were reduced on the filters using 100 μl 10 mM TCEP (77720, Bond-Breaker™ TCEP solution, Thermo Scientific) in 50 mM TEAB at 55 °C for 45 min followed by a 10 min spin at $14,000 \times g$, 20 °C and free cysteine residues were modified using 100 μl freshly prepared 15 mM iodoacetamide (IAA, Sigma-Aldrich) in 50 mM TEAB and incubated for 20 min at room temperature in the dark. The IAA solution was removed by washing with 10% acetonitrile (ACN) in 50 mM TEAB followed by centrifugation and filters transferred to new LoBind Eppendorf tubes. Tryptic digestion was performed by adding 1.6 μg of trypsin (V5111, Promega, sequencing grade modified trypsin) in 40 μl 10% ACN in 50 mM TEAB and incubated at 37 °C under humid conditions. Next day digested peptides were collected after spinning and then rinsing the filters with 60 μL 10% acetonitrile in 50 mM TEAB followed by a final centrifugation at $14,000 \times g$, which collected all tryptic peptides in the LoBind tube.

An equal amount (54 μg, determined by Pierce Quantitative Fluorometric peptide assay, 23275, Thermo Scientific) of peptides from each sample was subjected to isobaric labeling using Tandem Mass Tag (TMT-10plex) reagents (90110, Lot RG234662, Thermo Fischer Scientific) according to the manufacturer's instructions. The labeled samples were combined into one pooled sample, concentrated using vacuum centrifugation and separated into eight fractions using Pierce™ High pH Reversed-Phase Peptide Fractionation Kit (84868, Thermo Scientific) according to the manufacturer's instructions for TMT-labeled peptides. After vacuum centrifugation of peptide fraction to dryness the peptides were resuspended in 0.2% Formic Acid (FA) in 3% ACN.

**Nano-liquid chromatography tandem mass spectrometry with Q Exactive™ HF**. The TMT-labeled peptide samples were analyzed with an Easy-nLC1200 liquid chromatography system combined with Q Exactive HF mass spectrometer (Thermo Scientific) using a 136 min gradient. The separation was performed using an Acclaim PepMap precolumn (75 μM ID by 20 mm) connected to a 75 μM by 150 mm analytical Easy Spray PepMap RSLC C18 column (2 μm particles, 100 Å° pore size; Thermo Scientific) using a gradient from 5% solvent B to 15% solvent B over

47 min, then up to 25% B the next 58 min and up to 50% B in 20 min followed by an increase to 98% solvent B for 1 min, and 98% solvent B for 9 min at a flow of 280 nL/min. Solvent A was 0.1% formic acid and solvent B was 80% acetonitrile, 0.1% formic acid. MS scans were performed at 120,000 resolution, *m/z* range 350–1400. MS/MS analysis was performed in a data-dependent experiment, with top 15 of the most intense doubly or multiply positive charged precursor ions selected. Precursor ions were isolated in the quadrupole with a 1.2 *m/z* isolation window and 0.2 *m/z* offset, with dynamic exclusion set to a duration of 30 seconds. Isolated precursor ions were subjected to collision induced dissociation (CID) at 32 collision energy (arbitrary unit) with a maximum injection time of 100 ms. Produced MS2 fragment ions were detected at 60,000 resolutions, with a fixed first mass of 120 *m/z* and a scan range of 200–2000 *m/z*.

**Proteomic data analysis of Q Exactive™ HF data.** The data files were merged for identification and relative quantification using Proteome Discoverer version 2.1.1.21 (Thermo Fisher Scientific). Swiss-Prot Human database was used for the database search, using the Mascot search engine v. 2.5.1 (Matrix Science, London, UK) with MS peptide tolerance of 6 ppm and fragment ion tolerance of 0.02 Da. Tryptic peptides were accepted with 1 missed cleavage and methionine oxidation was set as a variable modification. Carbamidomethyl on cysteines and TMT on peptide N-termini and on lysine side chains were set as fixed modifications. Percolator was used for PSM validation with the strict FDR threshold of 1%. Quantification was performed in Proteome Discoverer 2.1.1.21. The TMT reporter ions were identified with 20 ppm mass tolerance in the MS2 spectra and the TMT reporter S/N values for each sample were normalized within Proteome Discoverer on the total peptide amount. Quantitative results were only based on unique peptide sequences with a co-isolation threshold of 50 and an average S/N threshold of 10 for the protein quantification.

**Sample preparation, peptide labeling and fractionation for Fusion™ Lumos™ Tribrid™ analysis.** Each sample was mixed with sodium dodecyl sulfate (SDS), triethylammonium bicarbonate (TEAB) and DL-dithiothreitol (DTT) to concentrations of 0.5% SDS, 50 mM TEAB, 100 mM DTT and incubated at 95 °C for 5 min for denaturation and reduction. The reduced samples were processed using the modified filter-aided sample preparation (FASP) method[70]. In short, the reduced samples were diluted to 1:4 by 8 M urea solution, transferred onto Nanosep 10k Omega filters (Pall Corporation, Port Washington, NY, USA) and washed repeatedly with 8 M urea and once with digestion buffer (0.5% sodium deoxycholate (SDC) in 50 mM TEAB). Free cysteine residues were modified using 10 mM methyl methanethiosulfonate (MMTS) solution in digestion buffer for 20 min at RT and the filters were washed twice with 100 µl of digestion buffer. One µg Pierce trypsin protease (MS Grade, Thermo Fisher Scientific) in digestion buffer was added and the samples were incubated at 37 °C for 3 h. An additional portion of trypsin was added and incubated overnight.

The peptides were collected by centrifugation and isobaric labeling was performed using Tandem Mass Tag (TMT-10plex) reagents (Thermo Fischer Scientific) according to the manufacturer's instructions. The labeled samples were combined into one pooled sample, concentrated using vacuum centrifugation, and SDC was removed by acidification with 10% TFA and subsequent centrifugation. The labeled pooled sample was treated with Pierce peptide desalting spin columns (Thermo Fischer Scientific) according to the manufacturer's instructions.

Each purified desalted sample was pre-fractionated into 40 primary fractions with basic reversed-phase chromatography (bRP-LC) using a Dionex Ultimate 3000 UPLC system (Thermo Fischer Scientific). Peptide separations were performed using a reversed-phase XBridge BEH C18 column (3.5 µm, 3.0 × 150 mm, Waters Corporation) and a linear gradient from 3% to 40% solvent B over 18 min followed by an increase to 100% solvent B over 5 min and 100% solvent B for 5 min at a flow of 400 µl/min. Solvent A was 10 mM ammonium formate buffer at pH 10.0 and solvent B was 90% acetonitrile, 10% 10 mM ammonium formate at pH 10.0. The fractions were concatenated into 20 fractions, dried and reconstituted in 3% acetonitrile, 0.2% formic acid.

**Nano-liquid chromatography tandem mass spectrometry with Fusion™ Lumos™ Tribrid™.** The fractions were analyzed on an orbitrap Fusion™ Lumos™ Tribrid™ mass spectrometer interfaced with Easy-nLC1200 liquid chromatography system (Thermo Fisher Scientific). Peptides were trapped on an Acclaim Pepmap 100 C18 trap column (100 µm × 2 cm, particle size 5 µm, Thermo Fischer Scientific) and separated on an in-house packed analytical column (75 µm × 35 cm, particle size 3 µm, Reprosil-Pur C18, Dr. Maisch) using a gradient from 5% solvent B to 33% solvent B over 77 min followed by an increase to 100% solvent B for 3 min, and 100% solvent B for 10 min at a flow of 300 nL/min. Solvent A was 0.2% formic acid and solvent B was 80% acetonitrile, 0.2% formic acid. MS scans were performed at 120,000 resolution, *m/z* range 375–1375. MS/MS analysis was performed in a data-dependent experiment, with top speed cycle of 3 s for the most intense doubly or multiply charged precursor ions. Precursor ions were isolated in the quadrupole with a 0.7 *m/z* isolation window, with dynamic exclusion set to 10 ppm and duration of 45 s. Isolated precursor ions were subjected to collision induced dissociation (CID) at 35 collision energy (arbitrary unit) with a maximum injection time of 50 ms. Produced MS2 fragment ions were detected in the ion trap followed by multinotch (simultaneous) isolation of the top 10 most abundant fragment ions for further fragmentation (MS3) by higher-energy collision dissociation (HCD) at 65% and detection in the Orbitrap at 50,000 resolutions, *m/z* range 100–500.

**Proteomic data analysis of Fusion™ Lumos™ Tribrid™ data.** The data files were merged for identification and relative quantification using Proteome Discoverer version 2.4 (Thermo Fisher Scientific). Swiss-Prot Human database was used for the database search, using the Mascot search engine v. 2.5.1 (Matrix Science, London, UK) with MS peptide tolerance of 5 ppm and fragment ion tolerance of 0.2 Da. Tryptic peptides were accepted with 0 missed cleavage and methionine oxidation was set as a variable modification. Cysteine methylthiolation and TMT on peptide N-termini and on lysine side chains were set as fixed modifications. Percolator was used for PSM validation with the strict FDR threshold of 1%. Quantification was performed in Proteome Discoverer 2.4. The TMT reporter ions were identified with 3 mmu mass tolerance in the MS3 HCD spectra and the TMT reporter S/N values for each sample were normalized within Proteome Discoverer 2.4 on the total peptide amount. Only the quantitative results for the unique peptide sequences with the minimum SPS match % of 40 and the average S/N above 10 were included for the protein quantification.

**Combined omics data analysis.** Proteomic data were compared to RNASeq results by pairing log2 fold-changes at the gene level and plotted in Fig. 6b. Data was plotted with R version 4.0.2 with ggplot2 version 3.3.5.

**Lipid droplet staining**
Liver spheroids in the culture compartments were fixed with 4% methanol-free paraformaldehyde (PFA; 28908, Thermo Scientific) at 4 °C overnight. On the following day, the spheroids were washed three times with PBS and then stored in PBS at 4 °C until use. Fixed spheroids were stained with 2 µM Nile Red (72485, Sigma-Aldrich) and 16 µM Hoechst 33342 (Invitrogen) in PBS. Samples were first incubated at 37 °C for 2 h, followed by an overnight incubation at RT. Next, the staining solution was removed, and the compartments were washed three times with PBS. Fluorescence imaging was performed using confocal laser scanning microscope (LSM880 Airyscan Zeiss) and image processing and reconstruction were carried out using ZEN 3.2 software (Zeiss).

## Glycogen staining

Liver compartments were washed with 0.1% BSA in PBS and the liver spheroids were detached from the bottom of the culture compartment using a sharp needle. The spheroids were transferred into 1.5 ml microtubes using wide-bore pipette tips for fixation using 4% methanol-free PFA at 4 °C overnight. The spheroids were then washed three times with PBS and stored at 4 °C until use. PAS staining to visualize the storage of glycogen was performed by Histocenter (Mölndal, Sweden). Briefly, after standard paraffin embedding and sectioning, the sections were sequentially treated with 0.5% periodic acid, water, Schiff reagent, water, Weigert's iron haematoxylin solution, water, hydrochloric acid, water, and 95% ethanol. Imaging was carried out using an inverted microscope (Axiovert 40 CFL, Zeiss).

## Cell proliferation analysis by EdU incorporation

We developed a method to quantify cell proliferation in pancreatic islets by using EdU incorporation, automated HT imaging, optical slicing, and automated image analysis (Supplementary Fig. 9a). To test robustness of the method, islets were cultured in Akura™ 96 Spheroid Microplate for 4 days, either in Human Islet Maintenance Medium (untreated control) or in the presence of 10 µM of the MST1 kinase inhibitor 4-(5-amino-6-(1-oxo-1,2,3,4-tetrahydroisoquinolin-6-yl)pyrazin-2-yl)-N-cyclopropyl-N-methylbenzenesulfonamide[71] (CAS 1396771-17-7) which was used as a positive control. To label proliferating cells, media were supplemented with 10 µM EdU. Donor for the robustness analysis study was a male, 45 years with BMI of 29.8 and HbA1c of 5.10%.

Fixation, permeabilization, and EdU staining were performed in Akura™ 96 Spheroid Microplates. The islets were fixed with 4% PFA at RT for 2 h, washed twice with 0.1% BSA in PBS, and permeabilized with 1x BD Perm/Wash buffer (554723, BD Biosciences) for 1 h at RT. Next, the islets were stained with Click-iT EdU reaction cocktail (C10638, Click-iT® Plus EdU Alexa Fluor® 555 Imaging Kit, Molecular Probes), for 2 h at RT in dark. After removal of the reaction cocktail, islets were washed once with 1x BD Perm/Wash buffer and transferred into Akura™ 384 Spheroid Microplate (CS-09-003-02, InSphero). Finally, a sorbitol-based clearing reagent Sca*l*e S4(0)[72] (40 (w/v)% D-(-)-Sorbitol (S3889, Sigma-Aldrich), 10(w/v)% Glycerol (G9012, Sigma-Aldrich), 4 M Urea (U0631, Sigma-Aldrich), 15–25(v/v)% DMSO) containing 3.0–3.9 µM SiR-DNA[73] (Spirochrome) for nuclear staining was added, and the plate was incubated overnight at RT. The plate was then centrifuged at $700 \times g$ for 1 min to remove bubbles and collect islets in the middle of wells and stored at 4 °C until imaging.

Images were acquired on a CellVoyager 7000 high-throughput spinning disc confocal microscope (Yokogawa). All microwells were first screened using a 10X 0.16NA objective at $2 \times 2$ binning (Supplementary Fig. 9b). A MATLAB-based Search First script (Wako Software Suite; Wako Automation) was used for automated detection of islet position in each micro-well. Then, high-resolution z-stacks of 200 µm from well bottom were acquired for each islet at its exact position, using a 40X 0.75NA objective at $2 \times 2$ binning in two fluorescence channels—EdU-positive nuclei (Click-iT EdU Alexa Flour 555; 561 nm laser) and nuclei (SiR-DNA far-red DNA stain; 640 nm laser) (Supplementary Fig. 9c). Using optical clearing in combination with 561 nm and 640 nm laser allowed for penetration of laser light and acquisition of fluorescent signal from throughout the islets, which usually have a diameter of 100–150 µm. Analysis of total number of nuclei and EdU-positive nuclei was performed using Columbus™ Image Data Storage and Analysis system (ver. 2.8.1, Perkin Elmer).

We observed that the percentage of EdU-positive cells is largely independent of optical sampling distance in the range of 0.4–20 µm (Supplementary Fig. 9d). Islets treated with the MST1 kinase inhibitor showed a significantly higher number of EdU-positive cells as compared to the untreated control islets (Supplementary Fig. 9e) demonstrating that the developed method can reliably separate different study groups.

In the pancreas-liver MPS, 10 µM EdU was added into co-culture medium for the last 5 days to label proliferating cells. After finishing a co-culture, islets were first transferred from chips into individual wells of an Akura™ 96 Spheroid Microplate followed by fixation, permeabilization, staining, and imaging as described above.

## Statistical analysis, evaluation of technical confounders and reproducibility analyses

GraphPad Prism software (Version 8) was used to plot the data and perform comparative analysis between the means of different conditions. For comparing two unpaired means of normal distributed data with homogenous variance, a two-tailed Student's $t$ test or a multiple $t$-test using the Holm–Sidak method (in case of several independent comparisons e.g., for comparing gene expression of multiple genes between two conditions) was performed. Normality was tested using the Shapiro–Wilk normality test and equality of variance was tested using the $F$-test. A $p$ value < 0.05 was considered statistically significant.

For comparing three or more means of normal distributed data with homogenous variance, a one-way ANOVA was performed. Normality was tested using the Shapiro–Wilk normality test and equality of variance was tested using the Brown–Forsythe test. Bonferroni's multiple comparison post hoc test was used to compare the means of several conditions to a control mean and Sidak's multiple comparison post hoc test was used to compare the mean of selected pairs of conditions. A $p$ value < 0.05 was considered statistically significant.

Fold changes of gene expression in Figs. 3 and 5 and GSIS data in Fig. 3e were log-transformed for normality. The area under the glucose and insulin GTT curves was calculated with GraphPad Prism using the trapezoidal method.

Evaluation of technical confounders and reproducibility was conducted using the R Statistical language (version 4.2.1). Mixed modeling was performed using the lmer() function from the lme4 package (version 1.1.34), with the Nelder-Mead optimizer to address convergence issues. Significance was tested through summary functions in the lmerTest package (version 3.1.3), using Satterthwaite's correction for degrees of freedom. Contrasts for estimated marginal means were computed using the emmeans package (version 1.8.2). All evaluated endpoints were log-transformed before analysis.

Data reproducibility metrics were calculated according to statistical methodology proposed by Schurdak et al.[41]. Briefly, we calculated reproducibility both across studies and within each study, for a given condition, by pooling all relevant values and applying the max coefficient of variation (max CV) and intra-class correlation coefficient (ICC). Max CV was computed by finding the maximum CV across all timepoints, where the CV is calculated as the ratio of the standard deviation to the mean, across all endpoint values at that timepoint:

$$\text{Max CV} = \text{Max}_t(\text{CV}_t) = \text{Max}_t\left(\frac{\hat{\sigma}_t}{\hat{\mu}_t}\right)$$

ICC procedure used was ICC2 from the R package *psych* (version 2.4.1), where each circuit was considered a "judge" and each timepoint a "target". Hence the formula is:

$$\text{ICC}(2,1) = \frac{\text{MS}_B - \text{MS}_E}{\text{MS}_B + (n_r - 1)\text{MS}_E + n_r(\text{MS}_J - \text{MS}_E)/n_c}$$

where $\text{MS}_B$ and $\text{MS}_J$ are the mean squares between targets and judges respectively, $\text{MS}_E$ is the residual mean square error, $n_r$ is the number of "judges", and $n_c$ is the total number of observations. Note that, in contrast to the mixed model approach above, data were not log-transformed before these procedures.

## Reporting summary

Further information on research design is available in the Nature Portfolio Reporting Summary linked to this article.

## Data availability

Transcriptomics data are publicly available on GEO (GSE249277). The mass spectrometry proteomics data have been deposited to the ProteomeXchange Consortium[74] via the PRIDE partner repository[75] with the dataset identifier PXD052854. All source data underlying the graphs presented in the main figures are available as Supplementary Data.

## Code availability

The code for data analysis, visualization, and mathematical modeling is publicly available on GitHub (https://github.com/belencasasgarcia/Insulin-resistance). The code for the mixed model statistical analysis is publicly available on Zenodo[76] (11545665).

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

## Acknowledgements

We thank S. Prill, A. Vildhede, B. Nugraha, N. Nguyen, K. Schimek, T. P. Tao, I. Durieux, S. Aljburi, and I. Rütschle for technical assistance as well as J. Dietzfelbinger for creating the graphical illustration of Fig. 1a–d. Part of the proteomic analysis was performed at the Proteomics Core Facility at Sahlgrenska Academy, University of Gothenburg. PAS staining was performed at Histocenter AB, Mölndal, Sweden. Sigrid Jusélius Foundation Postdoctoral Fellowship grant (L.K.V.); Swedish Foundation for Strategic Research grant ITM17-0245, Center for Industrial Information technology (CENIIT) 15.09 (G.C.); The Knut and Alice Wallenberg Foundation, SciLifeLab National COVID-19 Research Program 2020.0182 (G.C.); H2020 project PRECISE4Q 777107 (G.C.); Swedish Fund for Research without Animal Experiments F2019-0010 Excellence Center at Linköping (ELLIIT) 2020-A12 (G.C.); VINNOVA Visual Sweden and 2020-04711 (G.C.); Swedish Research Council, 2018-05418 and 2018-03319 (G.C.).

## Author contributions

Conceptualization: S.R., L.K.V., and T.B.A.; Methodology: S.R., K.P.K., C.W.H., E.M., C.Ä., T.B.A., and L.K.V.; Software: B.C.; Investigation: S.R., B.C., K.P.K., C.W.H., L.U.M., E.M., M.C., S.F.H., and L.K.V.; Formal analysis: F.K., L.L., and J.C.; Visualization: S.R., B.C., and L.K.V.; Supervision: L.K.V., T.B.A., C.Ä., U.M., G.C., P.G., and R.J.L.; Writing—original draft: S.R., B.C., and L.K.V.; Writing—review & editing: S.R., B.C., K.P.K., C.W.H., L.U.M., E.M., F.K., M.C., S.F.H., L.L., J.C., R.J.L., C.Ä., U.M., P.G., G.C., T.B.A., and L.K.V.

## Competing interests

K.P.K., C.W.H., L.M., E.M., F.K., M.C., S.F.H., L.L., J.C., R.J.L., P.G., C.Ä., T.B.A., and L.K.V. are employees or previous employees of AstraZeneca. U.M. is a founder, CSO, and shareholder of TissUse GmbH, which

commercializes MPS platforms. All other authors declare they have no competing interests.
