## [Peer Review File · Communications Biology]

Reviewers' comments:

Reviewer #1 (Remarks to the Author):

Vilen et al. report a new MPS system to study the bi-directional communication between liver hepatocytes and an in-vitro model of pancreatic islets. Their approach has been tested in two different laboratories with tissues stemming from multiple donors. The MPS technology used has a long track record and reliability.

The study represents an important stepping stone in modeling multiorgan human physiology in the context of systemic glucose metabolism.

The rationale, approach, and rigor of analysis are appropriate.

Please consider the following suggestions>

1. Diabetes, as noted in the text, is a complex pathology involving a plethora of different cell types and signaling paths. While the paper clearly shows expected responses from the system when challenged, it still only models a small portion of what we consider diabetes. Aside from missing tissues, tissue-resident and circulating immune cell populations play a significant role in liver-islet cross-talk and development of type II diabetes. It would be advisable for the title and conclusion to reflect the reductionist approach and focus on the interaction between hepatocytes and islets and glucose homeostasis rather than diabetes in general. The title also mentions "diseased" which one would expect means the cells and tissues were derived from patient populations. In the case of this study, certain pathological states were modeled, and given the lack of direct patient comparison "diseased" is too ambiguous.

2. I would like to remind the authors that insulin and its role were discovered in animal models. Also the vast majority of what we know about multiorgan signaling in connection to metabolism and diabetes were discovered in KO animal models. This gave rise to today's most prescribed and efficient drugs to maintain glucose control. It would be appropriate to reflect this in the paper rather than striking an all-out dismissive tone. You made the point for the need of human models to study new human-specific modalities. I would expand on that and further explore the precision-medicine aspects of MPS.

3. In the same spirit as above, please address the various shortcomings of the study and model as part of the discussion.

Reviewer #2 (Remarks to the Author):

The objective of the article Diseased human pancreas and liver microphysiological system for preclinical diabetes research is to develop a MPS with liver and pancreatic islets to disease model T2D. By modulating the glucose and hydrocortisone levels of the media the authors generated the

healthy and T2D diseased conditions. The authors evaluated reproducibility of the MPS model across multiple donors and repeatability across different laboratory. Finally, a new therapeutic target has been identified to induce islet proliferation.

The overall premise of the article, in developing MPS model for T2D is highly significant and the TissUse platform has been previously used and applied for that purpose. The current article is a follow up on the previous publication and has some new aspects in this study.

Overall the manuscript presents plethora of different analysis, ranging from computational modeling to biochemical characterization and further transcriptomic and proteomic analysis, which is a clear strength of this work. The extent of characterization is well appreciated. However, there are some major concerns specific conclusions drawn in the manuscript, as detailed below.

Overall the manuscript is poorly written and presented in a confusing way which makes it very difficult to read and follow the logic. As written the manuscript feels discordant and often the conclusions are not clearly supported by the presented data. It reads highly discordant and seems like multiple parallel and somewhat unrelated studies stitched together without a coherent story or specific outcome. Hence the article will need significant rewriting to justify the study and explain things in a logical and coherent manner.

One concern is the use of glucocorticoid hydrocortisone (HCT) which this work is trying to optimize for appropriate organ function in the MPS model. It is misleading to generalize it as T2D disease model since this MPS model is relevant to only glucocorticoid induced diabetes. While in the results section the authors have specified this aspect, the title, abstract, introduction and discussion is referring to general T2D, which is misleading and will be misinterpreted by the audience. At all points it needs to specify glucocorticoid induced diabetes, and motivate the work in the intro from that angle as well.

The initial section of the results is written as if a continuation of a previous manuscript which makes it very difficult to read and follow. The beginning of the results with reference to Fig 1 is not obvious if that is new data or reproduction of previously reported result? There is reference to aspects without any data supporting that but instead a reference to previous work (for eg: normoglycemia not restoring GSIS). If that is indeed prior work, it is best to include in the Introduction to clearly motivate this current work by highlighting limitations and shortcomings of previous work.

The first part of the manuscript is focused on establishing the healthy and diseased (T2D) condition in the coupled liver-islet system, which in essence the authors calibrate the HCT levels to establish. While the effect on liver is characterized in detail, the effect on islet is characterized only by insulin secretion and GSIS. While that is the primary organ function, it is reasonable to include (as an example) on/off-chip images (immunostaining) to demonstrate the phenotype of the islets in the healthy and diseased condition.

One of the conclusion in the manuscript is in the identification of IL1R2 as the primary inducer of pancreatic islet proliferation. The presented data however is correlative and the authors have not

established causative role to definitively draw this conclusion. Also the controls are not always appropriate either. In Fig 6A static culture is not the best control since it assumes that flow or on-chip culture has no effect on the islets. The ideal control would be mono-culture of the islets under identical condition as the co-culture but without the liver tissue.

The main point of 6A is that under healthy (low HCT) condition the co-culture induces more proliferation than static culture. This difference is lost in the disease condition. But under the co-culture in the MPS model is there difference in proliferation between healthy and disease condition? It is not obvious why the comparison with the static culture is relevant – since the disease modeling is happening entirely in the MPS.

Fig 6C is convincing that IL1R2 is increasing with time in the diseased co-culture condition, but the healthy condition has low levels of IL1R2. In order to link IL1R2 to cell proliferation the authors have presented static and standalone islet cultures data which clearly shows that IL1R2 promotes islet proliferation at low levels but not at high levels. However this is not adequate to conclude that IL1R2 is the liver secreted factor in the co-culture affecting cell proliferation. The disconnect here is in the effect of HCT, since low HCT (healthy co-culture) itself promotes significant proliferation (positive control of Fig 6D) it is not obvious how the authors are decoupling the effect of HCT-induced proliferation from the effect of IL1R2? In essence Fig 6 and all studies related to IL1R2 is conducted and presented in a confusing way without appropriate control and incorrect comparisons which, as presented, does not support the effect of IL1R2 on islet proliferation, as concluded in the manuscript.

A more clearer presentation of the control media conditions and a stronger justification of the conclusions is required to claim IL1R2 is at all playing a role here. If neutralizing IL1R2 under healthy condition reduces proliferation, it will indicate a causative effect. However, this also indicates the healthy condition set up in this system does not capture the healthy patient condition. But captures the compensatory pre-diabetic phase of the disease where islets are trying to increase mass in order to compensate for the IR stage.

The genomic and proteomic results are not presented in details. The only information included is the overlay of RNAseq and proteomic analysis to highlight IL1R2.

The authors have mentioned :

“We evaluated the model in two laboratories and observed low inter-experimental and inter laboratory variation.” This is a critical point and a great strength of the manuscript, since reproducibility is a key point to the success of MPS models. However it is not obvious to this reviewer which data is supporting this statement and what statistical analysis has been performed to quantify this variability. The results section does not mention which results are from which lab. This needs to be clearly stated and properly supported by statistical analysis to demonstrate the claim.

Reviewer #3 (Remarks to the Author):

Rigal et al. present a manuscript on a very relevant topic and employ an innovative combination of in silico and in vitro model to address it. The work addresses the important challenge of using a common media for connecting different tissues while assuring its metabolic stability. The claim that it is a disease model is not supported by the data as 14 days corticoid exposure is an acute stimulation and not a chronic disease state. Corticoid concentration is a well-known disruptor of energy metabolism with a known impact in insulin resistance. Furthermore, the IL1R2 is not sufficiently explored in this light. A further critical point is that insulin sensitivity of the liver is not measured and would be critical for the claim. Gene expression, PAS, lipid content as well as ketone bodies concentration at constant 10 nM insulin concentrations are not sufficient to determine the liver as insulin resistant.

The paper should be reframed in the sense of showing the importance of lower glucose and corticoid concentrations for a healthy and physiological relevant culture. Alternatively, new data demonstrating response to insulin in the liver depending on the GSIS in the pancreas should be included (e.g. AKT phosphorylation, foxO1 translocation, changes in endogenous glucose production by the liver).

Furthermore, a number of major concerns should be addressed before publication:

Results section

The structure needs improvement – there is a lack of a clear differentiation between introduction/results/discussions that makes the reading hard to follow. Its often not clear in the results section with which figure/data are the authors referring to. A major restructuring of the results and discussion sections is therefore required and the results section should be limited to presenting the results and explaining experimental design choices.

Example (lines 239 to 241): In summary, the reduction of HCT concentration to a physiological level improved the glucose-stimulated insulin secretion showing that the effect of a high HCT concentration is in line with the beta-cell failure observed in patients suffering from steroid diabetes(22).

Is this what the authors see in their data or are discussing data from reference 22?

Discussion

The discussion is confusing and split between results and discussion section. Relevant study design choices are not discussed in the light of literature and the relevance of low glucose and low HTC conditions for co-culture is not discussed, which in my opinion is the core data and most

relevant for the field. This is essential for metabolic multi-organ studies.

Figures

All figures require improvement to include the following details: clear figure title, level of detail, day of data collection, from single or co-culture, what is the color code and definition of studies and data points (a data point is a spheroid or a chip? A study is a donor or an independently run experiment?). Which data was collected from which laboratory?

Specific comments:

- Line 193 to 200: The authors mention that 5nM HCT does not differ from control. Why was then 10 nM chosen instead of 5 nM?

- Fig 3C – why was data from a different day in culture used for validation of the in silico? How is it confirmed that the cell culture is the same at day 13 as at day 3 or 9?

- Fig 3 – the figure panels are not ordered, and important information is missing, e. g. is A islets alone or connected to liver? How is hepatic control over insulin levels in the system accounted for or discussed? How does the fmol/2h translate into concentration? Where is the concentration measured? And what is the total system volume? For clarity, the concentration could be mentioned in the results or discussion sections.

- Fig 4 – it is not clear what the color code of panels A and B is. The figure caption is very confusing and needs to be re-written for clarity.

- Fig 5 E – in which nutritional conditions are the ketone bodies secreted?

- Fig 6. The data is presented in a very confusing way. Its not clear what is liver derived and what is externally controlled. Panel D is on which day? Is the hypothesis that the liver secretes IL1R2 that controls hepatic proliferation? How is it demonstrated that it indeed comes from the liver? Figure 6C should include a liver single culture. The authors mention no pancreatic secretion of IL-1R2 but what if the liver is producing it at higher rates in the co-culture than in liver alone?

- Fig 6 A and C – more pancreatic proliferation in healthy condition but lower IL1R2?

- Fig 6. B The multi-omics data is from 3 studies for RNA and 4 for proteomics – how are the donors matched here? How do you distinguish between real effects vs pancreatic/liver donor match? Do the data from 3 RNAseq datasets match 3 of the 4 used for proteomics? If not and no rationale, this needs clarification. Moreover, IL1R2 occurs as membrane-bound and soluble form. RNA data does not distinguish this, but proteomics does. Authors should discuss this.

Methods:

- The authors should add a section in experimental design defining what is a study, what is a readout independent assay and which/how many donors are in each study. A table with available donor information should also be included.
- Ratio of liver to pancreas of 4:1 spheroid number – how was this ratio determined?
- What is the total volume of the system and how was the media in the connection circuit exchanged? If the media is exclusively exchanged in the compartments, how do the authors account for this effect on the measured concentrations, e. g. of insulin? It is very important to also know the insulin concentration. It is not possible to compare with the provided information, but it is essential for replicating the work or to support other studies. Additionally, it is not clear where is the media was sampled for analysis – in the connection channel or in each tissue compartment?
- IL-1R2 treatment was performed in pancreatic compartment but the data is presented as liver effect in the pancreatic cells and from a single donor.
- Line 507 - Glucose tolerance test – please mention which methods and companies used to measure insulin – it's a key information that should not be only cited.

POINT-BY-POINT RESPONSE TO REVIEWERS (Rigal et al.)

Reviewer 1

- *Vilen et al. report a new MPS system to study the bi-directional communication between liver hepatocytes and an in-vitro model of pancreatic islets. Their approach has been tested in two different laboratories with tissues stemming from multiple donors. The MPS technology used has a long track record and reliability.*

The study represents an important stepping stone in modeling multiorgan human physiology in the context of systemic glucose metabolism.

The rationale, approach, and rigor of analysis are appropriate.

Please consider the following suggestions.

We thank the Reviewer for valuable feedback, raising important points, and acknowledging that our “*approach has been tested in two different laboratories*” and that the study “*represents an important stepping stone in modeling multiorgan human physiology in the context of systemic glucose metabolism.*” Our responses to the remaining concerns are described below. We refer to line numbers in the ‘Clean’ version of the revised manuscript. The version with Track Changes is also available. Modified figures are at the end of this document.

1.

- *“Diabetes, as noted in the text, is a complex pathology involving a plethora of different cell types and signaling paths. While the paper clearly shows expected responses from the system when challenged, it still only models a small portion of what we consider diabetes. Aside from missing tissues, tissue-resident and circulating immune cell populations play a significant role in liver-islet cross-talk and development of type II diabetes. It would be advisable for the title and conclusion to reflect the reductionist approach and focus on the interaction between hepatocytes and islets and glucose homeostasis rather than diabetes in general. The title also mentions “diseased” which one would expect means the cells and tissues were derived from patient populations. In the case of this study, certain pathological states were modeled, and given the lack of direct patient comparison “diseased” is too ambiguous.”*

To respond to these valid concerns, we have done following changes in the manuscript to acknowledge that our model is indeed limited to liver-pancreas axis and that other organ models and immune cells would add value.

In the discussion, we have added following sentences:

Lines 533-534: “Our current MPS is limited to modelling the liver-pancreas axis – a portion of insulin resistance which is a complex multi-organ disease.”

Lines 546-547: “Additionally, incorporating immune cells into the multi-organ MPS would further increase its physiological relevance⁵⁴.”

Throughout the manuscript, we now define the hyperglycemic high-cortisone condition as ‘glucocorticoid-induced diabetes’ and do not any longer generalize to diabetes or type 2 diabetes. We have kept the terms ‘healthy’ and ‘diseased’ chips when describing data from normoglycemic low-

cortisone condition ('healthy') and hyperglycemic high-cortisone ('diseased') after we have described the terms (lines 254-258).

We have changed the title to "Normoglycemia and physiological cortisone level maintain glucose homeostasis in a pancreas-liver microphysiological system".

2.

- *"I would like to remind the authors that insulin and its role were discovered in animal models. Also the vast majority of what we know about multiorgan signaling in connection to metabolism and diabetes were discovered in KO animal models. This gave rise to today's most prescribed and efficient drugs to maintain glucose control. It would be appropriate to reflect this in the paper rather than striking an all-out dismissive tone. You made the point for the need of human models to study new human-specific modalities. I would expand on that and further explore the precision-medicine aspects of MPS."*

This is fair and helpful criticism. We therefore have included sentences in the Introduction and Discussion to acknowledge the key role of animal models in the discovery of anti-diabetic drugs and their current role as a main experimental diabetes model.

Lines 84-87: "Historically, animal models have paved the way for discovering anti-diabetic treatments and they still serve as the main experimental model of diabetes allowing studies of the complex disease pathophysiology and multi-organ interactions⁶."

Lines 471-477: "Preclinical diabetes studies rely on animal models because the standard *in vitro* single-cell or single-organ cultures cannot replicate organ-to-organ crosstalk that is essential for multi-organ disorders. Furthermore, animal models allow for investigating multi-scale mechanisms that span across different timescales and biological levels, ranging from whole-body to intracellular⁴¹. Therefore, they are well suited to study key mechanisms related to metabolic disorders such as body-fat distribution, systemic glucose metabolism or brain control over metabolic fluxes⁶."

We have also added a new part with new references into the Discussion on how MPS could enable personalized medicine and expanded our point on using MPS for studying new human-specific modalities.

Lines 548-551: "Furthermore, the use of hiPSC-derived organ models could reflect the highly heterogeneous disease progression and patient-specific pathophysiology. Indeed, MPS with patient-derived cells hold a great potential for personalized medicine, including disease modelling of rare genetic diseases and selecting personalized drug treatments⁵⁵."

Lines 551-560: "Since the described pancreas-liver MPS is composed of only human cells, the model should also enable studies with new drug modalities such as oligonucleotide therapeutics. Generally, oligonucleotide therapeutics are specific for human gene sequence with limited homology to non-clinical species. Transgenic mice models or parallel development of species-specific oligonucleotide therapeutics are used to overcome the challenge with non-human preclinical models⁹. Recently, kidney MPS platform has been used to assess safety profiles of antisense oligonucleotides^{49,50}. Our pancreas-liver MPS could be used for efficacy and safety studies, but also to study targeting efficiency and non-target effects of oligonucleotide therapeutics."

3.

- *“In the same spirit as above, please address the various shortcomings of the study and model as part of the discussion.”*

We have revised the entire Discussion and replied to the specific concerns mentioned above. Additionally, we have included following other shortcomings of the study and limitations in the experimental design in the Discussion:

Lines 442-442: “However, we did not normalize the albumin secretion to total protein amount so the effect of cell proliferation cannot be excluded.”

Lines 447-449: “To confirm insulin resistance development in the liver spheroids, a functional readout showing an increased endogenous glucose production by the liver could be included in future studies.”

Lines 494-495: “More studies are needed to better understand the role of IL-1R2-mediated islet proliferation in the pancreas-liver MPS,”

Lines 497-502: “Finally, it is important to note that all studied co-culture media, including the healthy medium, induced proliferation compared to the islet maintenance in the medium provided by the islet supplier. Rather than reflecting a healthy pancreas, the islet phenotype in the described healthy medium might represent an initial pre-diabetic stage in which islets are trying to compensate by increasing their mass³¹. This increase in the basal proliferation might also be caused by the fetal bovine serum (FBS) used in the chip medium. FBS is widely used in cell culture to support cell growth and proliferation. In the future, we will investigate if a serum-free medium would reduce the basal islet proliferation in the pancreas-liver MPS to obtain a pancreas model that fully reflects the healthy pancreas.”

Lines 531-531: “To accommodate studies on inter-donor variability also for the liver part, we are currently developing a pancreas-liver MPS with primary human hepatocytes.”

Reviewer 2

- *The objective of the article Diseased human pancreas and liver microphysiological system for preclinical diabetes research is to develop a MPS with liver and pancreatic islets to disease model T2D. By modulating the glucose and hydrocortisone levels of the media the authors generated the healthy and T2D diseased conditions. The authors evaluated reproducibility of the MPS model across multiple donors and repeatability across different laboratory. Finally, a new therapeutic target has been identified to induce islet proliferation.*

The overall premise of the article, in developing MPS model for T2D is highly significant and the TissUse platform has been previously used and applied for that purpose. The current article is a follow up on the previous publication and has some new aspects in this study.

Overall the manuscript presents plethora of different analysis, ranging from computational modeling to biochemical characterization and further transcriptomic and proteomic analysis, which is a clear strength of this work. The extent of characterization is well appreciated. However, there are some major concerns specific conclusions drawn in the manuscript, as detailed below.

We thank the Reviewer for valuable and helpful feedback, raising important topics, and for acknowledging that we have “*evaluated reproducibility of the MPS model across multiple donors and repeatability across different laboratories*” and that “*developing MPS model for T2D is highly significant*”. Our responses to the concerns are described below. We refer to line numbers in the ‘Clean’ version of the revised manuscript. The version with Track Changes is also available. Modified figures are at the end of this document.

1.

- *Overall the manuscript is poorly written and presented in a confusing way which makes it very difficult to read and follow the logic. As written the manuscript feels discordant and often the conclusions are not clearly supported by the presented data. It reads highly discordant and seems like multiple parallel and somewhat unrelated studies stitched together without a coherent story or specific outcome. Hence the article will need significant rewriting to justify the study and explain things in a logical and coherent manner.*

This is fair criticism and we have substantially revised the manuscript and rewritten major parts of the Introduction, Results, and Discussion to explain the work and findings in a more logical and coherent manner.

2.

- *One concern is the use of glucocorticoid hydrocortisone (HCT) which this work is trying to optimize for appropriate organ function in the MPS model. It is misleading to generalize it as T2D disease model since this MPS model is relevant to only glucocorticoid induced diabetes. While in the results section the authors have specified this aspect, the title, abstract, introduction and discussion is referring to general T2D, which is misleading and will be misinterpreted by the audience. At all points it needs to specify glucocorticoid induced diabetes, and motivate the work in the intro from that angle as well.*

This is very helpful criticism. Throughout the manuscript, we now define the hyperglycemic high-cortisone condition as ‘glucocorticoid-induced diabetes’ and do not any longer generalize to type 2 diabetes. We have kept the terms ‘healthy’ and ‘diseased’ chips when describing data from normoglycemic low-cortisone condition (‘healthy’) and hyperglycemic high-cortisone (‘diseased’)

after we have described the terms (lines 254-258). Moreover, we decided to lift the importance of the defined physiological culture condition and, thus, we have changed the title to “Normoglycemia and physiological cortisone level maintain glucose homeostasis in a pancreas-liver microphysiological system”. This title is in line with being specific about the role of glucocorticoids.

3.

- *The initial section of the results is written as if a continuation of a previous manuscript which makes it very difficult to read and follow. The beginning of the results with reference to Fig 1 is not obvious if that is new data or reproduction of previously reported result? There is reference to aspects without any data supporting that but instead a reference to previous work (for eg: normoglycemia not restoring GSIS). If that is indeed prior work, it is best to include in the Introduction to clearly motivate this current work by highlighting limitations and shortcomings of previous work.*

We agree that in the previous version the findings from the previous papers were not well enough presented and that it was not logical to place previous findings in the Results sections. Now, we have rewritten the Introduction where we mention all previous findings (lines 105-116) and then discuss these in the Discussion.

4.

- *The first part of the manuscript is focused on establishing the healthy and diseased (T2D) condition in the coupled liver-islet system, which in essence the authors calibrate the HCT levels to establish. While the effect on liver is characterized in detail, the effect on islet is characterized only by insulin secretion and GSIS. While that is the primary organ function, it is reasonable to include (as an example) on/off-chip images (immunostaining) to demonstrate the phenotype of the islets in the healthy and diseased condition.*

In this work, we did not include protein expression analysis (immunostaining) of pancreatic islets as we had done this previously in our first pancreas-liver MPS paper (Bauer et al. 2017 Sci Rep) where we showed insulin staining for beta cells and glucagon staining for alpha cells. A careful protein expression analysis by staining where the aim is to quantify the expression (to show difference between healthy and disease condition) would have required too large number of islets (there are 10 islets per compartment) and thus chip replicates and we made a choice to use the islets for functional readouts including off-chip GSIS and EDU proliferation assay. However, we plan to do immunostaining in the future studies.

5.

- *One of the conclusion in the manuscript is in the identification of IL1R2 as the primary inducer of pancreatic islet proliferation. The presented data however is correlative and the authors have not established causative role to definitively draw this conclusion. Also the controls are not always appropriate either. In Fig 6A static culture is not the best control since it assumes that flow or on-chip culture has no effect on the islets. The ideal control would be mono-culture of the islets under identical condition as the co-culture but without the liver tissue. The main point of 6A is that under healthy (low HCT) condition the co-culture induces more proliferation than static culture. This difference is lost in the disease condition. But under the co-culture in the MPS model is there difference in proliferation between healthy and disease condition? It is not obvious why the comparison with the static culture is relevant – since the disease modeling is happening entirely in the MPS.*

This is fair and helpful criticism. We have now more clearly written that in this exploratory work, we use combined omics to profile soluble proteins that could explain our observation that the islets proliferate more in the chip co-culture vs standard static monoculture, only as a proof-of-concept to demonstrate that the multi-organ MPS can be used to identify new therapeutic proteins. We do not state that the IL-1R2 is the primary inducer of islet proliferation and we acknowledge that much more studies are needed to better understand the role of IL-1R2 in the liver-islet communication. However, such studies were not the focus of this work. We have now more clearly described this in the Discussion.

Lines 492-497: “These results suggest that chip-born IL-1R2 may be one of the factors impacting the islet proliferation in the healthy pancreas-liver co-culture. More studies are needed to better understand the role of IL-1R2-mediated islet proliferation in the pancreas-liver MPS, but our data on chip-born factors that modulate the islet proliferation demonstrates, as a proof-of-concept, that multi-organ MPS can be used to find and study new targets and therapeutic proteins.”

We chose to compare the proliferation of co-cultured to static islets, because we saw a difference in the GSIS (Fig. 3E). Therefore, we investigated if the observed increase in GSIS in the hyperglycaemic condition might be correlated to an increased proliferation. In Fig. 6A, we can see that also in co-culture, there is a significant difference in proliferation between the healthy and diseased medium. We have now added this into the figure ($p=0.0004$) and in the Results:

Lines 304-305: “Additionally, the chip co-cultured islets are more proliferative in the healthy condition than in the disease condition.”

Although the flow can partly affect the islet proliferation and we agree that the proposed on-chip islet monocultures would be the ideal control, the difference between the healthy and diseased on-chip conditions shows that there is flow-independent proliferation (Fig 6A). As the proliferation in statically monocultured islets is not induced by the healthy medium, we believe that the difference between healthy and diseased co-cultured islets is most likely caused by a crosstalk with the liver model. We have modified the Discussion and do not call the IL-1R2 liver-derived but talk about chip-born factors as mentioned above (lines 492-497).

6.

- *Fig 6C is convincing that IL1R2 is increasing with time in the diseased co-culture condition, but the healthy condition has low levels of IL1R2. In order to link IL1R2 to cell proliferation the authors have presented static and standalone islet cultures data which clearly shows that IL1R2 promotes islet proliferation at low levels but not at high levels. However this is not adequate to conclude that IL1R2 is the liver secreted factor in the co-culture affecting cell proliferation. The disconnect here is in the effect of HCT, since low HCT (healthy co-culture) itself promotes significant proliferation (positive control of Fig 6D) it is not obvious how the authors are decoupling the effect of HCT-induced proliferation from the effect of IL1R2? In essence Fig 6 and all studies related to IL1R2 is conducted and presented in a confusing way without appropriate control and incorrect comparisons which, as presented, does not support the effect of IL1R2 on islet proliferation, as concluded in the manuscript.*

It is correct that the healthy co-culture medium itself promotes proliferation as compared to the islets maintained in the medium provided by the cell supplier. We have added following sentences to the Discussion:

Lines 497-506: “Finally, it is important to note that all studied co-culture media, including the healthy medium, induced proliferation compared to the maintenance in the medium provided by the islet supplier. Rather than reflecting a healthy pancreas, the islet phenotype in the

described healthy medium might represent an initial pre-diabetic stage in which islets are trying to compensate by increasing their mass³¹. This increase in the basal proliferation might also be caused by the fetal bovine serum (FBS) used in the chip medium. FBS is widely used in cell culture to support cell growth and proliferation. In the future, we will investigate if a serum-free medium would reduce the basal islet proliferation in the pancreas-liver MPS to obtain a pancreas model that fully reflects the healthy pancreas.”

We carried out the IL-1R2 study with monocultured islets in the suppliers medium to decouple the effect of the co-culture medium itself. Even if we can see an increased islet proliferation at the low dose (0.3 ng/mL) which is the same level of IL-1R2 measured from the healthy chips, we do not state in the manuscript that the IL-1R2 is the primary inducer of islet proliferation as stated above in Q5. Also, we have added the following sentence in the Results:

Lines 326-328: We cultured the islets in the medium provided by the islet supplier as the chip co-culture media induce proliferation (Fig. S7), and we wanted to decouple the effect of IL-1R2 from any medium-induced proliferation.”

7.

- *A more clearer presentation of the control media conditions and a stronger justification of the conclusions is required to claim IL1R2 is at all playing a role here. If neutralizing IL1R2 under healthy condition reduces proliferation, it will indicate a causative effect. However, this also indicates the healthy condition set up in this system does not capture the healthy patient condition. But captures the compensatory pre-diabetic phase of the disease where islets are trying to increase mass in order to compensate for the IR stage.*

As described above, we have added a sentence (lines 326-328) to Results that we used islet maintenance medium as a basal medium and this served as an untreated control. In the Materials and Methods and in Fig. 6 legend, respectively, we write:

Lines 666-669: “Islets cultured in insulin-free chip co-culture medium with 10 nM hydrocortisone and 11 mM glucose (Low HCT - HG condition) served as a positive control since this medium was shown to significantly induce islet proliferation (Fig. S7).”

Lines 1673-632: “IL-1R2 stimulates cell proliferation at low dose (0.3 ng/mL) but not at the high dose (30 ng/mL) in islets monocultured in static condition in culture medium provided by the islet manufacturer.”

Line 1677: “Positive control: hyperglycemic low-HCT co-culture medium.”

As mentioned above, we have included this suggestion on compensatory effect during pre-diabetic phase now in the Discussion (lines 499-506).

8.

- *The genomic and proteomic results are not presented in details. The only information included is the overlay of RNAseq and proteomic analysis to highlight IL1R2.*

In this work, we used the combined omics to profile soluble proteins and to demonstrate, as a proof-of-concept, that the multi-organ MPS can be used to identify new therapeutic proteins. We understand that the data sets from RNA-Seq and proteomics would allow us to do more in-depth analysis but that

was not our focus of the work. We provide both data sets to the scientific community by depositing them into public repositories.

In the Materials and Methods, we include a description of the combined data analysis:

Lines 1162-1164: “Proteomic data was compared to RNASeq results by pairing log₂ fold-changes at the gene level and plotted in Fig. 6B. Data was plotted with R version 4.0.2 with ggplot2 version 3.3.5.”

9.

- *The authors have mentioned :*
“We evaluated the model in two laboratories and observed low inter-experimental and inter laboratory variation.” This is a critical point and a great strength of the manuscript, since reproducibility is a key point to the success of MPS models. However it is not obvious to this reviewer which data is supporting this statement and what statistical analysis has been performed to quantify this variability. The results section does not mention which results are from which lab. This needs to be clearly stated and properly supported by statistical analysis to demonstrate the claim.

This is relevant and helpful criticism. In the revised version, we include new data from statistical analysis and have written a new Results chapter “Reproducibility of the pancreas-liver MPS” (lines 338-379), added new Fig. 7 with data from the statistical analysis, and discuss the topic on variability in *in vitro* systems and our results in the Discussion (lines 507-526; 566-567). Additionally, we have included Table S1 to give information about the experimental design (e.g., including N numbers in each study and in which laboratory the study was conducted) and Table S2 in which we list all pancreatic islet donors used in each study. Finally, we have added information to each figure legend in which laboratory the study was done.

Reviewer 3

- *Rigal et al. present a manuscript on a very relevant topic and employ an innovative combination of in silico and in vitro model to address it. The work addresses the important challenge of using a common media for connecting different tissues while assuring its metabolic stability. The claim that it is a disease model is not supported by the data as 14 days corticoid exposure is an acute stimulation and not a chronic disease state. Corticoid concentration is a well-known disruptor of energy metabolism with a known impact in insulin resistance. Furthermore, the IL1R2 is not sufficiently explored in this light. A further critical point is that insulin sensitivity of the liver is not measured and would be critical for the claim. Gene expression, PAS, lipid content as well as ketone bodies concentration at constant 10 nM insulin concentrations are not sufficient to determine the liver as insulin resistant.*

The paper should be reframed in the sense of showing the importance of lower glucose and corticoid concentrations for a healthy and physiological relevant culture. Alternatively, new data demonstrating response to insulin in the liver depending on the GSIS in the pancreas should be included (e.g. AKT phosphorylation, foxO1 translocation, changes in endogenous glucose production by the liver).

Furthermore, a number of major concerns should be addressed before publication:

We thank the Reviewer for detailed and helpful feedback, raising important points, and acknowledging that we “*present a manuscript on a very relevant topic and employ an innovative combination of in silico and in vitro model to address it*”. “*The work addresses the important challenge of using a common media for connecting different tissues while assuring its metabolic stability.*” Our responses to the concerns are described below. We refer to line numbers in the ‘Clean’ version of the revised manuscript. The version with Track Changes is also available. Modified figures are at the end of this document.

The Reviewer’s criticism is fair and helpful. Throughout the manuscript, we now define the hyperglycemic high-cortisone condition as ‘glucocorticoid-induced diabetes’ and do not any longer generalize to type 2 diabetes. We have kept the terms ‘healthy’ and ‘diseased’ chips when describing data from normoglycemic low-cortisone condition (‘healthy’) and hyperglycemic high-cortisone (‘diseased’) after we have described the terms (lines 254-258).

We agree that our work on IL-1R2 is not broadly exploring its role in the islet-liver communication and organ functionality. However, in this manuscript we used combined omics to profile soluble proteins that could explain our observation that the islets proliferate more in the chip co-culture compared to the standard static monoculture, only as a proof-of-concept to demonstrate that the multi-organ MPS can be used to identify new therapeutic proteins. We agree that much more studies are needed to better understand the role of IL-1R2, but this is not the focus of this current work. We have now more clearly described this in the Discussion.

Lines 492-497: “These results suggest that a chip-born IL-1R2 may be one of the factors impacting the islet proliferation in the healthy pancreas-liver co-culture. More studies are needed to better understand the role of IL-1R2-mediated islet proliferation in the pancreas-liver MPS, but our data on chip-born factors that modulate the islet proliferation demonstrates, as a proof-of-concept, that multi-organ MPS can be used to find and study new targets and therapeutic proteins.”

In our earlier work, the first description of the islet-liver MPS (Bauer et al. 2017 Sci Rep), we characterized the insulin sensitivity of liver spheroids by measuring AKT phosphorylation before the chip cultures. However, even then we were not able to carry out this measurement from chip cultures since the needed cellular material for the assay is too high and thus impossible to get a result from one circuit. Also, pooling cellular material from multiple circuits would hinder doing other assays

(staining, qPCR) which we have considered more valuable. However, we agree that more functional assays focusing on the liver insulin resistance should be carried out in the future. One such a study would be analysis of endogenous glucose production by using C13-labelled gluconeogenic substrates. Therefore, we have added the following sentence to the Discussion:

Lines 447-449: “To further confirm insulin resistance development in the liver spheroids, a functional readout showing an increased endogenous glucose production by the liver could be included in future studies.”

In response to the Reviewer’s important point on reframing the paper, we have indeed lifted the description of the defined physiological culture condition and, thus, have changed the title to “Normoglycemia and physiological cortisone level maintain glucose homeostasis in a pancreas-liver microphysiological system”. We have included a new paragraph in the Discussion:

Lines 462-470: “Identification of the components driving the glucose dysregulation in the pancreas-liver MPS allowed us to select co-culture conditions that maintain the glucose homeostasis over two weeks. Defining physiological culture conditions, 5.5 mM glucose and 10 nM HCT, significantly increases the value of the model and allows studies on disease progression and mechanisms when the results from a diseased co-culture can be compared to a healthy condition. Furthermore, the described healthy condition can be used as a control group when studying other metabolic disorders or development of insulin resistance by other disease mediators than hyperglycemia and glucocorticoids, such as free fatty acids, fructose, and cytokines⁴⁵.”

Results section

2.

- *The structure needs improvement – there is a lack of a clear differentiation between introduction/results/discussions that makes the reading hard to follow. Its often not clear in the results section with which figure/data are the authors referring to. A major restructuring of the results and discussion sections is therefore required and the results section should be limited to presenting the results and explaining experimental design choices.*

Example (lines 239 to 241): In summary, the reduction of HCT concentration to a physiological level improved the glucose-stimulated insulin secretion showing that the effect of a high HCT concentration is in line with the beta-cell failure observed in patients suffering from steroid diabetes(22).

Is this what the authors see in their data or are discussing data from reference 22?

We have substantially revised the manuscript and rewritten major parts of the Introduction, Results, and Discussion to explain the work and findings in a more logical and coherent manner. We agree that in the previous version the Results section included unnecessary discussion which we now have moved into the Discussion section. The example (lines 239-241 in the previous version) given by the Reviewer has been removed from the Results.

Discussion

3.

- *The discussion is confusing and split between results and discussion section. Relevant study design choices are not discussed in the light of literature and the relevance of low glucose and*

low HTC conditions for co-culture is not discussed, which in my opinion is the core data and most relevant for the field. This is essential for metabolic multi-organ studies.

As mentioned above, we have substantially rewritten major parts of the Results and Discussion. We have moved several references justifying our study design choices from the results section to the discussion to improve readability for the reader. For example:

Lines 396-398: "As previously reported, we initially suspected that the driver of the glucose dysregulation in the pancreas-liver n the pancreas-liver MPS^{12,14} is hyperglycemia since it is a known inducer of insulin resistance^{15,16}."

Lines 428-430: "In the liver, glucocorticoids have been reported to increase glucose production via gluconeogenesis⁴¹ and to promote hepatic lipid accumulation (steatosis)²⁶ which is suspected to induce insulin resistance⁴²."

As also mentioned above, we have lifted description of the low glucose and low HCT condition in the revised version. This healthy condition is now discussed in lines 462-470.

Figures

4.

- *All figures require improvement to include the following details: clear figure title, level of detail, day of data collection, from single or co-culture, what is the color code and definition of studies and data points (a data point is a spheroid or a chip? A study is a donor or an independently run experiment?). Which data was collected from which laboratory?*

This is helpful feedback, and we have modified all figures according to this comment. Additionally, we have included Table S1 to give information about the experimental design (e.g., including N numbers in each study and in which laboratory the study was conducted) and Table S2 in which we list all pancreatic islet donors used in each study.

Specific comments:

5.

- *Line 193 to 200: The authors mention that 5nM HCT does not differ from control. Why was then 10 nM chosen instead of 5 nM?*

We chose to use 10 nM HCT concentration instead of the 5 nM since 10 nM is within the physiological plasma concentration range which is reported to be about 5.5-39 nM. We have added the following sentence to the Results:

Lines 192-193: "We set the tested low HCT concentration to 10 nM to be within the physiological plasma concentration range."

6.

- *Fig 3C – why was data from a different day in culture used for validation of the in silico? How is it confirmed that the cell culture is the same at day 13 as at day 3 or 9?*

We understand that the Reviewer refers to Fig 2C instead of Fig 3C (albumin production). The *in silico* model we use in this manuscript, has been described in the Casas *et al.* 2022 PLOS Comput Biol

paper. In that previous work, we carried out seven independent pancreas-liver MPS studies with seven islet donors to build the mathematical model. We did show that the model can predict glucose and insulin responses at the end of the culture (days 13-15) when feeding in GTT data from earlier timepoints of the study. The model was shown to correctly predict glucose and insulin responses also from a glycemic condition (2.8 mM glucose) – a condition that had never been used to train the model. In this work in the hypothesis testing experiment (Fig 2), we feed the mathematical model with glucose and insulin data from the GTT assay carried out on days 1-3 and 7-9 to calibrate the *in silico* model for donor-specific insulin secretion capacity. By doing this calibration for both models corresponding to the H1 and H2 hypotheses, we can then generate a prediction of the glucose and insulin responses on days 13-15 when an insulin dose is injected in the MPS. How these predictions compare to the new experimental data will determine which hypothesis can be rejected. There are two major reasons for using validation data between days 13-15: first, validation needs to be performed using data not included for calibration and second, the behaviour of interest for distinguishing between hypothesis H1 and H2 is expected to be observed at the end of the co-culture (days 13-15) when glucose dysregulation is present. It is important to note that this *in silico* model calibration was done ‘in real-time’ as the *in vitro* MPS study was ongoing at the same time.

7.

- *Fig 3 – the figure panels are not ordered, and important information is missing, e. g. is A islets alone or connected to liver? How is hepatic control over insulin levels in the system accounted for or discussed? How does the fmol/2h translate into concentration? Where is the concentration measured? And what is the total system volume? For clarity, the concentration could be mentioned in the results or discussion sections.*

We have reorganized the figures A-D in the panel. We have included more detailed description in the figure legend stating that the study was done with monocultured islets in static condition.

Lines 1569-1571: “Hydrocortisone (HCT)-concentration-dependent inhibition of glucose-stimulated insulin secretion (GSIS) of islets cultured in static monoculture in normoglycemic co-culture medium.”

We have previously demonstrated the hepatic control over the insulin levels by comparing pancreas-liver co-cultures to mono-pancreas and mono-liver cultures. In our Bauer *et al.* (2017) Sci Rep paper, we showed that insulin secreted into the circulation stimulated glucose uptake by the liver spheroids, and when the glucose concentration fell, the insulin secretion subsided.

In the GSIS assay, we report the insulin secretion as “fmol/2h” because data in the form of “amount/time” could be more easily compared across different assay systems. We have also understood that this form “amount/time” is the most common way to report GSIS data. We have described the assay volumes in the materials and methods allowing readers to calculate the concentrations.

8.

- *Fig 4 – it is not clear what the color code of panels A and B is. The figure caption is very confusing and needs to be re-written for clarity.*

We have added the colour codes for panels A and B and rewritten the legend.

9.

- *Fig 5 E – in which nutritional conditions are the ketone bodies secreted?*

The ketone body secretion was studied from the Healthy (low HCT low glucose) and Diseased (high HCT and high glucose) conditions. This information has now been added to the Fig. 5E.

10.

- *Fig 6. The data is presented in a very confusing way. Its not clear what is liver derived and what is externally controlled. Panel D is on which day? Is the hypothesis that the liver secretes IL1R2 that controls hepatic proliferation? How is it demonstrated that it indeed comes from the liver? Figure 6C should include a liver single culture. The authors mention no pancreatic section of IL-1R2 but what if the liver is producing it at higher rates in the co-culture than in liver alone?*

This is fair and helpful criticism. As mentioned above (Q1), we do not study the role of IL-1R2 in this work in detail and we emphasize the exploratory nature of the IL-1R2 work in the current version of the manuscript:

Lines 492-497: “These results suggest that a chip-born IL-1R2 may be one of the factors impacting the islet proliferation in the healthy pancreas-liver co-culture. More studies are needed to better understand the role of IL-1R2-mediated islet proliferation in the pancreas-liver MPS, but our data on chip-born factors that modulate the islet proliferation demonstrates, as a proof-of-concept, that multi-organ MPS can be used to find and study new targets and therapeutic proteins.”

In the revised version, we have modified the Fig. 6 legend, now entitled: “Assessment of soluble factors promoting islet proliferation on chip”.

We understand that Reviewer means ‘islet proliferation’, not actually ‘hepatic proliferation’ (*Is the hypothesis that the liver secretes IL1R2 that controls hepatic proliferation?*) We have not mentioned hepatic proliferation in this work. We do hypothesise that liver secretes IL-1R2 that controls the islet proliferation. Although the flow (external effect) can partly affect the islet proliferation, we can see flow-independent proliferation on chip. On chip, there is difference in the proliferation between the healthy and diseased conditions (Fig 6A). As the proliferation in statically mono-cultured islets is not induced by the healthy medium, we believe that the difference between healthy and diseased co-cultured islets is most likely caused by a crosstalk with the liver model. To study this, we performed the combined omics analysis (RNA-Seq of liver spheroids and proteomics of co-culture supernatant). In this work, we follow up with one of the omics hit proteins, IL-1R2 that we show is stimulating the islet proliferation (Fig. 6D). In this work, we wanted to treat the islet with IL-1R2 concentrations measured in the co-culture (Fig 6C), and thus we have not included single-liver controls. It is a good suggestion to use single-liver culture as a control to study the role of IL-1R2 more in detail.

The IL-1R2 work was carried out using single islets in static culture in the cell supplier’s medium to disconnect the effect of IL-1R2 from the co-culture medium. We had described the experimental design of the IL-1R2 exposure in the Methods, but we have now added details about the culture time (total IL-1R2 exposure time 16 days) to the figure legend.

11.

- *Fig 6 A and C – more pancreatic proliferation in healthy condition but lower IL1R2?*

Our data from two independent studies performed in different laboratories show that the healthy condition stimulates islet proliferation and the measured IL-1R2 concentrations are lower compared to the diseased condition. In Fig. 6A, we can see that in co-culture, there is a significant difference in proliferation between the healthy and diseased medium. We have now added this into the figure ($p=0.0004$) and in the Results:

Lines 304-305: “Additionally, the chip co-cultured islets are more proliferative in the healthy condition than in the disease condition.”

We had mentioned already in the previous version: “Interestingly, it has been reported that low, but not high, IL-1beta concentration has beneficial effects on islet functionality^{44,45}. Therefore, it may be possible that the low IL-1R2 concentration in the healthy condition might have reduced the IL-1beta concentration to a beneficial range while the high IL-1R2 concentrations resulted in ineffectively low IL-1beta concentrations.” In the current version, this is moved from Results into the Discussion (lines 488-492)

12.

- *Fig 6. B The multi-omics data is from 3 studies for RNA and 4 for proteomics – how are the donors matched here? How do you distinguish between real effects vs pancreatic/liver donor match? Do the data from 3 RNAseq datasets match 3 of the 4 used for proteomics? If not and no rational, this needs clarification. Moreover, IL1R2 occurs as membrane-bound and soluble form. RNA data does not distinguish this, but proteomics does. Authors should discuss this.*

We have now included Table S1 to give information about the experimental design (e.g., including N numbers in each study and in which laboratory the study was conducted) and Table S2 in which we list all pancreatic islet donors used in each study. In the combined omics studies, every donor was exposed to high and low glucose concentrations and analysis was performed with donor as covariate to remove chip effects for both RNASeq and proteomics data. Same donors were used in RNASeq and proteomics data analysis, except for 1 additional donor in the proteomics data. We have added this description in the Fig. 6 legend:

Lines 1667-1670: “Two studies were performed at TissUse (matching donors in RNA-Seq and proteomics analysis) and two studies were performed at AstraZeneca (one combined RNA-Seq and proteomics analysis with a matching donor and one additional donor in proteomics analysis.”

We thank the reviewer for reminding about the two forms of IL-1R2 and we now included both in the Results and in the Discussion that in this work we show that the soluble form of IL-1R2 has an impact on islet proliferation. We agree that RNA-Seq cannot differentiate between the soluble and membrane-bound IL-1R2 levels while the proteomics can. There is, however, no real conflict in the RNA-seq and proteomics data: RNA-Seq shows 1.9-fold upregulation and proteomics data 1.5-fold upregulation. The form of IL-1R2 that we detect in the media by proteomics (in the absence of cell death as shown by stable albumin secretion, Fig. 3C) is the soluble form, and we used soluble form (human recombinant protein) to treat monocultured islets (Fig 6D).

In the revised version, we have included the following sentence and added that we detect soluble form with the proteomics analysis and that we treated the static monocultured islets with recombinant human IL-1R2 (soluble form).

Lines 311-312: “IL-1R2 exists both as soluble and membrane-bound decoy receptor, a competitive inhibitor preventing IL-1beta signalling.”

Methods:

13.

- *The authors should add a section in experimental design defining what is a study, what is a readout independent assay and which/how many donors are in each study. A table with available donor information should also be included.*

This is fair and helpful criticism. As mentioned above, we have included Table S1 and S2 for details about experimental design and donors used. Also, we have added information to each figure legend in which laboratory the study was done.

14.

- *Ratio of liver to pancreas of 4:1 spheroid number – how was this ratio determined?*

We have described the pancreas-liver MPS first time in Bauer et al. (2017) Sci Rep in which we described the scaling factor: “Ten pancreatic islet microtissues were placed in the compartment upstream of a chamber containing 40 liver spheroids (Fig. 1C). This is equivalent to a factor of 100,000 present in a normal human pancreas (estimated to contain on the order of one million pancreatic islets). The number of liver spheroids represents a similar fraction of a normal human liver.” More precisely, we based our liver scaling on the total number of hepatocytes. This number however differs greatly between individuals. Assuming that the human liver contains around $114\text{-}164 \times 10^6$ hepatocytes per gram of liver (Sohlenius-Sternbeck, *Toxicol In Vitro* 2006, <https://doi.org:10.1016/j.tiv.2006.06.003>) and male livers weigh between 838 and 2584 g (Molina & DiMaio *Am J Forensic Med Pathol* 2012, <https://doi.org:10.1097/PAF.0b013e31823d29ad>), the total number of hepatocyte for an individual ranges between $96\text{-}424 \times 10^9$ cells. For downscaling of our liver model, we decided to use the lower end of this range to match the scaling of the pancreas (scaling factor 100,000). Therefore, our liver model has 40 liver spheroids with 24,000 HepaRG cells and 1000 stellate cells each, resulting in 960,000 hepatocytes. Nevertheless, we are aware that using the lowest hepatocytes number might result in a scaling mismatch in the translation to human. However, we have shown in our previous publication on MPS combined with *in silico* modelling (Casas et al. 2022 PLO Comput Biol) that such mismatches can be accounted for using computational modelling approach.

15.

- *What is the total volume of the system and how was the media in the connection circuit exchanged? If the media is exclusively exchanged in the compartments, how do the authors account for this effect on the measured concentrations, e. g. of insulin? It is very important to also know the insulin concentration. It is not possible to compare with the provided information, but it is essential for replicating the work or to support other studies. Additionally, it is not clear where is the media was sampled for analysis – in the connection channel or in each tissue compartment?*

The total volume of the system (in each circuit) is 605 μl . We described this in the previous version in the methods but now added this information to Fig 1. legend as well for clarity. At every medium change, the medium in compartments (300 μl + 300 μl) is exchanged completely while the residual volume 5 μl remains in the microfluidic channels. Insulin concentration in the circuit is also measured at the beginning of each GTT assay: 0 h sample is taken from the circuit immediately after the medium change and this data is included in the GTT insulin graphs.

Lines 624-626: “Three days before insertion of the organ models, the chips were prepared for cultivation by replacing the storage buffer with 300 μ L co-culture medium in each culture compartment (total volume per circulation was 605 μ L).”

Lines 675-679: “Briefly, we exchanged the co-culture medium in both culture compartments with a co-culture medium containing 11 mM glucose (-300 μ L, +315 μ L) and collected 15 μ L of supernatant samples at 0, 8, 24, and 48 h to monitor glucose and insulin concentrations. To obtain sufficient sample volumes for the analysis, samples from the liver and islet compartment were pooled.”

Lines 1536-1538: “Each circuit contains 605 μ l of co-culture medium (5 μ l in the microfluidic channels and 300 μ l in each culture compartment). Medium is exchanged by replacing the total volume in each culture compartment.”

16.

- *IL-1R2 treatment was performed in pancreatic compartment but the data is presented as liver effect in the pancreatic cells and from a single donor.*

Since we had observed higher islet proliferation on chip compared to the static cultures (Fig 6A), we hypothesised that liver compartment can secrete proteins into the circulating co-culture medium that stimulate the proliferation. To study this, we performed the combined omics analysis (RNA-Seq of liver spheroids and proteomics of co-culture supernatant). In this work, we follow up with one of the omics hit proteins, IL-1R2 that we show is stimulating the islet proliferation. The IL-1R2 work was carried out using single islets in static culture in the cell supplier’s medium to disconnect the effect of IL-1R2 from the co-culture medium. We agree that more studies and more donors are needed to better understand the role of IL-1R2, and we now have included this in the Discussion.

Lines 492-497: These results suggest that chip-born IL-1R2 may be one of the factors impacting the islet proliferation in the healthy pancreas-liver co-culture. More studies are needed to better understand the role of IL-1R2-mediated islet proliferation in the pancreas-liver MPS, but our data on chip-born factors that modulate the islet proliferation demonstrates, as a proof-of-concept, that multi-organ MPS can be used to find and study new targets and therapeutic proteins.”

17.

- *Line 507 - Glucose tolerance test – please mention which methods and companies used to measure insulin – it’s a key information that should not be only cited.*

This was described in the methods chapter ‘Analysis of soluble markers’ and we have now also mentioned this in the ‘Glucose tolerance test’ chapter.

Lines 703-705: “Samples taken during the GTT were analysed for glucose (1070-500, Stanbio Laboratory) and insulin (10-1113-10, Mercodia) according to the manufacturer’s instructions.”

Line 682: “see Analysis of soluble markers”

Revised figures:

Fig. 2. Colour codes added for D and E. Visual improvement in B.

Fig. 3 Figures reorganized and p-values added.

Fig. 4. Colour codes added to A and B. P-values added to C.

Fig 5. Colour codes added to E and p-values added to A, B, C, and E.

Fig. 6 Colour codes added to C. P-values added to A and D.

Fig. 7. New figure.

Reviewers' comments:

Reviewer #4 (Remarks to the Author):

Overall, the article "Normoglycemia and physiological cortisone level maintain glucose homeostasis in a pancreas-liver microphysiological system" will be a valuable resource to the MPS community, especially for modeling complex metabolic disorders like MASLD-T2DM whose pathogenesis involves a complex interplay between multiple organ systems. Using MPS to study complex disease is an iterative process, and this work represents an important step in more fully implementing the use of MPS as both a mechanistic and drug discovery tool. The authors have thoughtfully and thoroughly addressed the reviewers' concerns and have significantly improved the quality and understand-ability of this work. This manuscript should be accepted for publication based on these revisions with the consideration of the final point described below:

Reviewer #2 (point 9) makes a critical point in asking for clarification of what statistical method(s) were used to assess to quantify the technical/experimental variability of the studies performed in this work. The authors responded by creating a new Figure 7 which partially answers reviewer 2's question; however, it seems that one element that is still lacking is an assessment of inter-study reproducibility among the 3 studies performed. Inter-study reproducibility should be assessed for the metrics associated with both Figure 7 (albumin and ketone bodies) as well as the metrics in Fig S11 (IL-1R2, GTT glucose, and GTT insulin). Inter-study reproducibility can be calculated using various statistical methods, and PubMed ID: 32211684 offers an informative guide for analyzing inter-study reproducibility in MPS studies. This work would be further strengthened by analyzing inter-study reproducibility in Figs 7 and S11 in this manner. Applying this type of analysis to human MPS studies is essential to further validating their use as drug development tools in both academia and industry.

POINT-BY-POINT RESPONSE TO REVIEWERS (Rigal et al.)

Reviewer 4

Overall, the article "Normoglycemia and physiological cortisone level maintain glucose homeostasis in a pancreas-liver microphysiological system" will be a valuable resource to the MPS community, especially for modeling complex metabolic disorders like MASLD-T2DM whose pathogenesis involves a complex interplay between between multiple organ systems. Using MPS to study complex disease is an iterative process, and this work represents an important step in more fully implementing the use of MPS as both a mechanistic and drug discovery tool. The authors have thoughtfully and thoroughly addressed the reviewers' concerns and have significantly improved the quality and understand-ability of this work. This manuscript should be accepted for publication based on these revisions with the consideration of the final point described below:

Reviewer #2 (point 9) makes a critical point in asking for clarification of what statistical method(s) were used to assess to quantify the technical/experimental variability of the studies performed in this work. The authors responded by creating a new Figure 7 which partially answers reviewer 2's question; however, it seems that one element that is still lacking is an assessment of inter-study reproducibility among the 3 studies performed. Inter-study reproducibility should be assessed for the metrics associated with both Figure 7 (albumin and ketone bodies) as well as the metrics in Fig S11 (IL-1R2, GTT glucose, and GTT insulin). Inter-study reproducibility can be calculated using various statistical methods, and PubMed ID: 32211684 offers an informative guide for analyzing inter-study reproducibility in MPS studies. This work would be further strengthened by analyzing inter-study reproducibility in Figs 7 and S11 in this manner. Applying this type of analysis to human MPS studies is essential to further validating their use as drug development tools in both academia and industry.

Answer:

We thank the Reviewer for valuable feedback and acknowledging that the described pancreas-liver MPS “will be a valuable resource to the MPS community”. We are thankful for the Reviewer for proposing further statistical analysis on the inter-study reproducibility. Our response to this suggestion is described below. New data tables (Tables 1 and 2) including new reproducibility metrics and new Fig S12 with line plots from individual circuits from all studies are at the end of this document. Additionally, we have improved the legend of Fig. 7 to better explain the statistical mixed model approach.

We have used the statistical methodology published by Schurdak *et al.* (PubMed ID: 32211684) to analyse reproducibility as advised by the Reviewer. We have done the following additions and modifications to the revised manuscript (here new text is marked in blue):

In the abstract:

Lines 54-55:

“This method was **reproducible** in two laboratories and was effective in multiple pancreatic islet donors.” (*Instead of an old version “This method was evaluated for reproducibility in two laboratories and was effective in multiple pancreatic islet donors.”*)

In the Results:

Lines 382-391:

“Finally, we assessed intra-study and inter-study reproducibility with a recently published statistical methodology⁴¹ based on intra-class correlation coefficient (ICC) and the maximum coefficient of variation (CV). In this analysis, the intra-study reproducibility of albumin, ketone bodies, and IL-1R2 measurements were classified as acceptable to excellent (Table 1). Reproducibility of glucose and insulin measurements during the GTT assay, the main readouts of the pancreas-liver model, were classified as excellent. The inter-study reproducibility in

both healthy and diseased condition was either acceptable or excellent for all other readouts except for ketone bodies that was classified as poor in the healthy condition (Table 2). Results from all on-chip readouts from each individual circuit in all studies are presented in Fig S12.”

In the Discussion:

Lines 541-552

“We also analysed the intra-study and inter-study reproducibility with a statistical methodology recently proposed by Schurdak et al.⁴¹. These analyses confirmed that the on-chip readouts have overall acceptable or excellent reproducibility within and across studies. Importantly, the key readout of the model, the GTT with glucose and insulin measurements, was classified with excellent reproducibility indicating high reliability of the model’s main context of use. The statistical analysis also revealed that the inter-study reproducibility of ketone body measurement was poor in the healthy condition. Similar to the findings reported by Schurdak et al.⁴¹, the poor reproducibility status was the result of one outlying study. Here, potential reasons could be either the cell source (difference between HepaRG batches) or the bioanalysis from fresh or frozen supernatants (study 1 and 2 fresh, study 3 frozen).”

Lines 592-593:

“We tested the pancreas-liver MPS in two laboratories and showed good ~~replicability and~~ intra-study and inter-study reproducibility.”

In the Materials and Methods:

Lines 1288-1302:

“Data reproducibility metrics were calculated according to statistical methodology proposed by Schurdak *et al*⁴¹. Briefly, we calculated reproducibility both across studies and within each study, for a given condition, by pooling all relevant values and applying the max coefficient of variation (max CV) and intra-class correlation coefficient (ICC). Max CV was computed by finding the maximum CV across all timepoints, where the CV is calculated as the ratio of the standard deviation to the mean, across all endpoint values at that timepoint:

$$Max CV = Max_t(CV_t) = Max_t\left(\frac{\hat{\sigma}_t}{\hat{\mu}_t}\right)$$

ICC procedure used was ICC2 from the R package *psych* (version 2.4.1), where each circuit was considered a “judge” and each timepoint a “target”. Hence the formula is:

$$ICC(2,1) = \frac{MS_B - MS_E}{MS_B + (n_r - 1)MS_E + n_r(MS_J - MS_E)/n_c}$$

where MS_B and MS_J are the mean squares between targets and judges respectively, MS_E is the residual mean square error, n_r is the number of “judges”, and n_c is the total number of observations. Note that, in contrast to the mixed model approach above, data were not log-transformed before these procedures.

New Tables and Figures added to the manuscript:

Table 1. Intra-study reproducibility of the pancreas-liver MPS. Statistical assessment of the reproducibility is based on intra-class correlation coefficient (ICC) and the maximum coefficient of variation (CV). Reproducibility is classified as “Excellent” if $\text{Max CV} \leq 5\%$ or $\text{ICC} \geq 0.8$; “Acceptable” if $5\% < \text{Max CV} < 15\%$ or $0.2 \leq \text{ICC} < 0.8$; or “Poor” if $\text{ICC} < 0.2$. Studies 1 and 2 were performed at TissUse and study 3 at AstraZeneca.

Healthy condition						
On-chip readout	Study	# of circuits	# of time points	Max CV	ICC	Reproducibility status
Albumin	1	10	7	25.2	0.805	Excellent (ICC)
	2	8	7	17.1	0.743	Acceptable (ICC)
	3	8	7	26.3	0.349	Acceptable (ICC)
Ketone bodies	1	8	2	7.6	0.422	Acceptable (ICC)
	2	8	2	13.2	0.464	Acceptable (ICC)
	3	8	2	15.4	0.355	Acceptable (ICC)
IL-1R2	2	4	3	46.3	0.969	Excellent (ICC)
	3	4	3	49.9	0.665	Acceptable (ICC)
GTT Insulin	1	4	4	9.4	0.908	Excellent (ICC)
	2	4	4	2.9	0.960	Excellent (ICC, CV)
GTT Glucose	1	4	4	20.0	0.953	Excellent (ICC)
	2	4	4	61.6	0.927	Excellent (ICC)
Diseased condition						
On-chip readout	Study	# of circuits	# of time points	Max CV	ICC	Reproducibility status
Albumin	1	6	7	36.5	0.581	Acceptable (ICC)
	2	4	7	37.1	0.479	Acceptable (ICC)
	3	4	7	20.7	0.890	Excellent (ICC)
Ketone bodies	1	6	2	16.3	0.843	Excellent (ICC)
	2	4	2	38.2	0.459	Acceptable (ICC)
	3	4	2	38.0	0.294	Acceptable (ICC)
IL-1R2	2	4	3	14.5	0.939	Excellent (ICC)
	3	4	3	7.3	0.977	Excellent (ICC)
GTT Insulin	1	4	4	3.5	0.724	Excellent (CV)
	2	4	4	5.1	0.882	Excellent (ICC)
GTT Glucose	1	4	4	50.6	0.905	Excellent (ICC)
	2	4	4	48.1	0.946	Excellent (ICC)

Table 2. Inter-study reproducibility of the pancreas-liver MPS. Statistical assessment of the reproducibility is based on the intra-class correlation coefficient (ICC) and the maximum coefficient of variation (CV). Reproducibility is classified as “Excellent” if $\text{Max CV} \leq 5\%$ or $\text{ICC} \geq 0.8$; “Acceptable” if $5\% < \text{Max CV} < 15\%$ or $0.2 \leq \text{ICC} < 0.8$; or “Poor” if $\text{ICC} < 0.2$. Studies 1 and 2 were performed at TissUse and study 3 at AstraZeneca.

Healthy condition						
On-chip readout	Studies	# of circuits	# of time points	Max CV	ICC	Reproducibility status
Albumin	1,2,3	26	7	34.6	0.277	Acceptable (ICC)
Ketone bodies	1,2,3	24	2	32.1	0	Poor (CV)
IL-1R2	2,3	8	3	54.5	0.739	Acceptable (ICC)
GTT Insulin	1,2	8	4	92.8	0.499	Acceptable (ICC)
GTT Glucose	1,2	8	4	7.1	0.861	Excellent (ICC)
Diseased condition						
On-chip readout	Studies	# of circuits	# of time points	Max CV	ICC	Reproducibility status
Albumin	1,2,3	14	7	40	0.553	Acceptable (ICC)
Ketone bodies	1,2,3	14	2	30	0.566	Acceptable (ICC)
IL-1R2	2,3	8	3	24	0.924	Excellent (ICC)
GTT Insulin	1,2	8	4	89.6	0.384	Acceptable (ICC)
GTT Glucose	1,2	8	4	4.8	0.753	Excellent (CV)

Figure S12.

On-chip readout results from individual circuits. Results presented as line plots from individual circuits from all studies for A) GTT glucose, B) GTT insulin, C) albumin secretion, D) ketone bodies, and E) IL-1R2. Studies 1 and 2 were performed at TissUse and study 3 at AstraZeneca.

REVIEWERS' COMMENTS:

Reviewer #4 (Remarks to the Author):

All comments were addressed thoroughly and thoughtfully, including the addition of new reproducibility analysis to address intra- and inter-study reproducibility. These analyses enhance both the veracity and interpretation of data, making it stronger and more practically relevant. This manuscript should now be accepted for publication. Very nice work.